Corrected: Author correction

# Dynamic ubiquitylation of Sox2 regulates proteostasis and governs neural progenitor cell differentiation

Chun-Ping Cui[1], Yuan Zhang[1], Chanjuan Wang[1], Fang Yuan[2,3], Hongchang Li[1], Yuying Yao[1], Yuhan Chen[1], Chunnan Li[1], Wenyi Wei[4], Cui Hua Liu [5], Fuchu He[1], Yan Liu [2,3] & Lingqiang Zhang[1]

Sox2 is a key transcriptional factor for maintaining pluripotency of stem cells. Sox2 deficiency causes neurodegeneration and impairs neurogenesis. Although the transcriptional regulation of Sox2 has been extensively studied, the mechanisms that control Sox2 protein turnover are yet to be clarified. Here we show that the RING-finger ubiquitin ligase complex CUL4A$^{DET1\text{-}COP1}$ and the deubiquitylase OTUD7B govern Sox2 protein stability during neural progenitor cells (NPCs) differentiation. Sox2 expression declines concordantly with OTUD7B and reciprocally with CUL4A and COP1 levels upon NPCs differentiation. COP1, as the substrate receptor, interacts directly with and ubiquitylates Sox2, while OTUD7B removes polyUb conjugates from Sox2 and increases its stability. COP1 knockdown stabilizes Sox2 and prevents differentiation, while OTUD7B knockdown destabilizes Sox2 and induces differentiation. Thus, CUL4A$^{DET1\text{-}COP1}$ and OTUD7B exert opposite roles in regulating Sox2 protein stability at the post-translational level, which represents a critical regulatory mechanism involved in the maintenance and differentiation of NPCs.

[1] State Key Laboratory of Proteomics, National Center of Protein Sciences (Beijing), Beijing Institute of Lifeomics, 100850 Beijing, China. [2] State Key Laboratory of Reproductive Medicine, Institute for Stem Cell and Neural Regeneration, School of Pharmacy, Nanjing Medical University, Nanjing, China. [3] Institute for Stem Cell and Regeneration, Chinese Academy of Science, 100101 Beijing, China. [4] Department of Pathology, Beth Israel Deaconess Medical Center, Harvard Medical School, Boston, USA. [5] CAS Key Laboratory of Pathogenic Microbiology and Immunology, Institute of Microbiology, Chinese Academy of Sciences, 100101 Beijing, China. These authors contributed equally: Chun-Ping Cui, Yuan Zhang, Chanjuan Wang, Fang Yuan. Correspondence and requests for materials should be addressed to Y.L. (email: yanliu@njmu.edu.cn) or to L.Z. (email: zhanglq@nic.bmi.ac.cn)

Neural progenitor/stem cells (NPCs) are present during the development of the central nervous system (CNS) and persist into adulthood in certain locations[1]. The balance between NPCs maintenance and differentiation is essential for supplying the brain with specific neural populations, both under physiological and pathological conditions. Transcription factors of stem cell are relevant to direct cell fate determination, and gaining insights into the regulatory machinery is critical for the control of NPCs identity, especially in guiding transitions between cell fates. The Sex determining region Y-box 2 (Sox2) is a key factor for maintaining NPCs and embryonic stem cell (ESC) pluripotency[1–4]. Sox2 encoding one of core transcriptional factors in cellular reprogramming is expressed at early stages of CNS development and marks NPCs[1–3]. Sox2 deficiency causes neurodegeneration and impairs neurogenesis[5–7]. So the molecular factors and mechanisms underlying Sox2 expression and activity regulation are critical for understanding the process of neurogenesis and neurodegeneration. The transcriptional regulation of Sox2 has been extensively documented[8,9], and the functional roles of phosphorylation[10], acetylation[11], SUMOylation[12], and methylation[13] of Sox2 in ESCs had been reported previously. In NPCs, however the mechanisms that stabilize Sox2 by post-translational modification (PTM) remain unknown.

The relative abundance and functional modifications of proteins are regulated by a complicated cellular machine, the ubiquitin-proteasome system (UPS) that specifically adds or removes away ubiquitin to or from the target proteins[14]. The specificity of the reaction is provided by the E3 ligase complex, which conjugates activated ubiquitin to the substrates. At the same time, the UPS is also regulated by a class of deubiquitylating enzymes responsible for removing ubiquitin conjugates from the substrates[14]. UPS pathway plays an essential role in regulation of pluripotency and cellular reprogramming[15] and furnished as many drug targets[16]. During ESCs differentiation, Sox2 undergoes proteasomal degradation[13,14]. Fang et al.[13] reported that SET domain-containing lysine methyltransferase 7 (Set7, also called SETD7) monomethylates Sox2 at K119, which induces Sox2 ubiquitylation and degradation. The homologous to E6-AP C-terminus (HECT)-type E3 ligase WW domain-containing protein 2 (WWP2) specifically interacts with K119-methylated Sox2 through its HECT domain to promote Sox2 ubiquitylation. In contrast, AKT1 (also known as protein kinase B) phosphorylates Sox2 at T118 and stabilizes Sox2 by antagonizing K119me by Set7 and vice versa. In mouse ESCs, AKT1 activity toward Sox2 is greater than that of Set7, leading to Sox2 stabilization and ESC maintenance[13]. Additionally, a recent study exhibited that Ub-conjugating enzyme E2S (Ube2S) mediates K11-linked polyubiquitin chain formation at the Sox2-K123 residue and reinforces the self-renewal and pluripotent state of mouse ES cells[17].

Here we show that the Cullin-RING finger ligase (CRL) complex CUL4A$^{DET1-COP1}$ and the deubiquitylase (DUB) OTUD7B/Cezanne-1 govern Sox2 protein stability during NPCs differentiation. Sox2 expression declines concordantly with OTUD7B and reciprocally with Cullin 4A (CUL4A) and constitutive photomorphogenic 1 (COP1, also known as RFWD2) protein levels upon NPCs differentiation. CUL4A$^{DET1-COP1}$ and OTU domain-containing protein 7B (OTUD7B) play roles in fining tune Sox2 stability by ubiquitylation or deubiquitylation, which represents a critical regulatory mechanism governing the maintenance and differentiation of NPCs and might be potential targets for the treatment of neural degenerative diseases.

## Results

**Sox2 is ubiquitylated during neuronal differentiation.** To examine the dynamics of Sox2 expression during NPC differentiation, human pluripotent stem cells were cultured in neural induction medium (NIM) and differentiated into neurospheres. Neurospheres were plated on matrigel-coated six-well plate for neuronal differentiation as described previously[18,19]. Sox2 mRNA level remained constant for the duration of the 9-day differentiation assay (Supplementary Fig. 1a), while Sox2 protein level gradually decreased with cell differentiation marked by TUJ1 (type III β-tubulin, a neuronal differentiation marker) (Fig. 1a, c). Then we treated NPCs with the protein synthesis inhibitor cycloheximide (CHX), and the half-life of Sox2 was significantly shortened after the induction of differentiation (Fig. 1a). Further, we examined the Sox2 ubiquitylation during NPCs differentiation. As shown in Fig. 1b, ubiquitylated Sox2 was markedly increased after the induction. These results suggested that Sox2 protein stability is fine-tuned for NPCs fate determination.

WWP2, Set7, and Ube2S have been reported to regulate Sox2 proteostasis by UPS in ESCs[13,17]. We wonder if these enzymes are involved in the regulation of Sox2 stability in NPCs. Results of immunoblot indicated that WWP2, the known E3 ligase for Sox2 in ESCs[13] is hardly detectable in NPCs and neurons (Supplementary Fig. 1b and c), suggesting that in NPCs, WWP2 seems not play as the major E3 ligase for Sox2 degradation. Additionally, we also determined the protein level of Set7 and Ube2S in human ESCs, NPCs and neurons, respectively. As shown in Supplementary Fig. 1c, expression of Set7 and Ube2S was not correlated with Sox2 or TUJ1 level, suggesting that there are other enzymes are involved in Sox2 ubiquitylation in NPCs.

**CUL4A contributes to Sox2 ubiquitylation in NPCs.** To identify the specific E3 ligase responsible for Sox2 in NPCs, we firstly screened six cullins (CUL1, 2, 3, 4A, 4B, and 5) that each assembles a multi-subunit CRL[20,21], the largest known class of ubiquitin ligases. The protein level of CUL4A, but not other cullins was upregulated during NPC differentiation and inversely correlated with Sox2 (Fig. 1c). CUL4 differs from other cullins in that it employs the WD40-like repeat-containing protein damaged DNA-binding protein 1 (DDB1) as its adaptor to link CUL4 and DDB1-CUL4-associated factors (DCAFs)[21]. DDB1 has been documented to regulate some of vital cellular pathways as an integral component of the CUL4 CRL[21,22]. To further confirm the effect of CUL4A E3 complex in NPCs, we examined Sox2 level in cells with knockdown of either CUL1, CUL2, CUL3, CUL4A, or DDB1. The results as shown in Fig. 1d showed that either CUL4A or DDB1 knockdown increased the Sox2 protein level markedly. These results underscored our speculation on CUL4A E3 ligase complex ubiquitylating Sox2 for degradation. Additionally, this finding suggested that DDB1 deficiency might inhibit NPC differentiation and lead to abnormal development of the nervous system. Consistently, Cang et al. reported that null mutation of DDB1 caused early embryonic lethality, while conditional inactivation of DDB1 in brain and lens led to neuronal and lens degeneration, brain hemorrhages, and neonatal death[23]. So the regulation of DDB1 on Sox2 might reveal a mechanism underlying DDB1 knockout leading to neuronal degeneration.

We further showed that CUL4A overexpression decreased Sox2 level (Fig. 1e, Supplementary Fig. 1d and e). The depletion of CUL4A remarkably suppressed Sox2 ubiquitylation (Fig. 1f), and its overexpression enhanced Sox2 ubiquitylation (Fig. 1g) after treatment with MG132, the proteasome inhibitor. Furthermore, we used CUL4A-deficient or CUL4B-deficient mouse embryonic fibroblasts (MEFs) to confirm the effect of CUL4A on Sox2 stability. The results showed that knockout of CUL4A, but not CUL4B, remarkably increased the level of Sox2 as well as that of E2F1 (Fig. 1h) which has been shown as a direct substrate of CUL4A$^{Cdt2}$ ligase[24]. In contrast, the level of KLF4, another

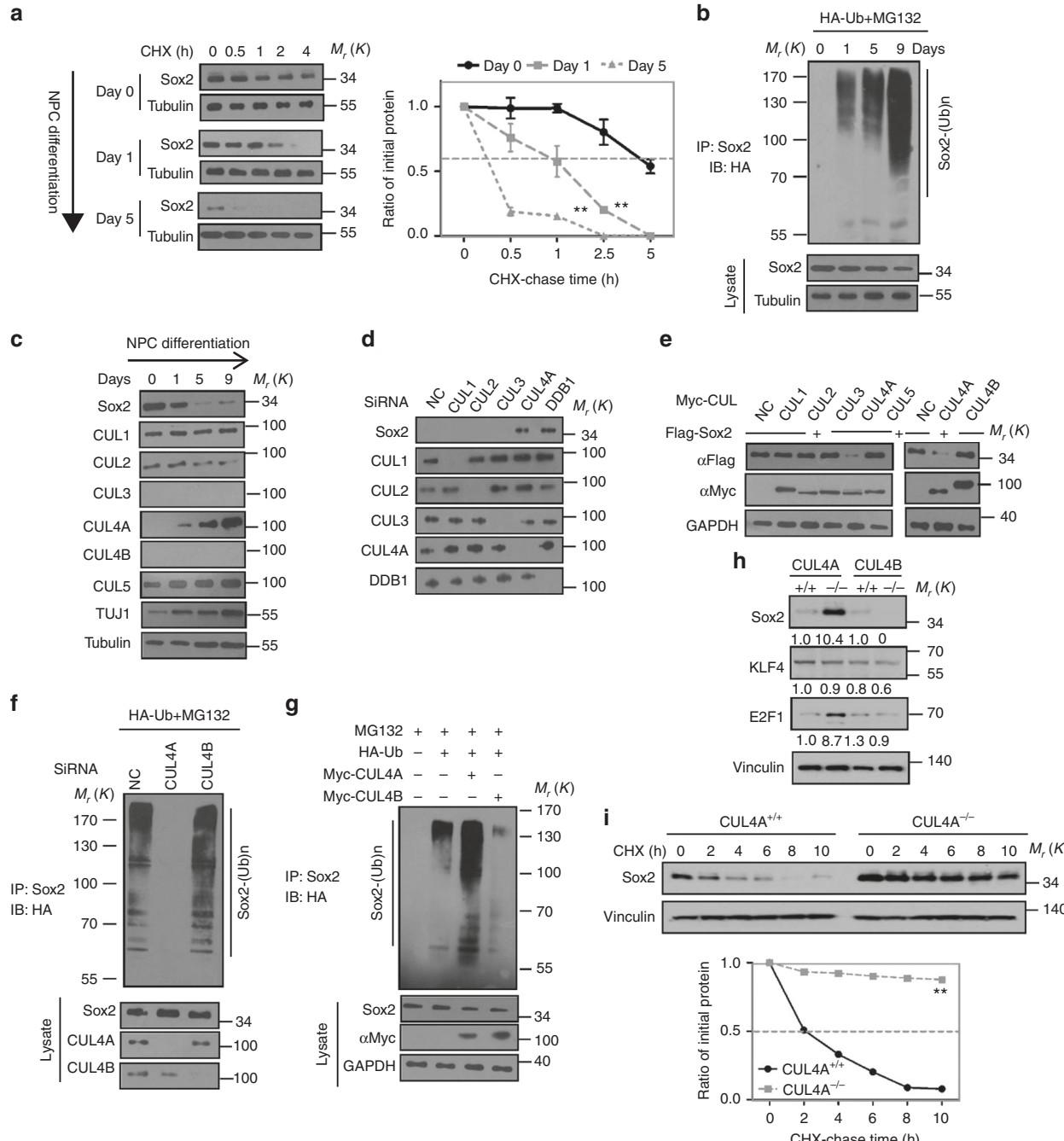

**Fig. 1** CUL4A contributes to Sox2 ubiquitylation in NPCs differentiation. **a** NPCs (neural stem/progenitor cells derived from human NPSCs, WA09) were cultured in neural induction medium (NIM) to undergo cellular differentiation for the indicated times. NPCs were treated with 10 μg per ml CHX, and collected at the indicated times for western blot. Quantification of Sox2 levels relative to tubulin is shown. Results are shown as mean ± s.d. $n = 3$ independent experiments. **$P < 0.01$, two-way ANOVA test. **b** NPCs transfected with HA-Ub were treated with MG132 for 8 h before collection. Sox2 was immunoprecipitated with anti-Sox2 antibody and immunoblotted with anti-HA antibody. **c** Immunoblotting of Sox2, CUL1, CUL2, CUL3, CUL4A, CUL4B, CUL5, and TUJ1 during NPCs differentiation. **d** Immunoblotting of Sox2 in HEK293T cells transfected with indicated siRNA. NC represents the empty vector control. **e** HEK293T cells were transfected with indicated constructs and the level of Flag-Sox2 was detected by immunoblotting. NC represents the empty vector control. **f, g** HA-Ub was co-transfected together with indicated siRNA (**f**) or constructs (**g**) into HEK293T cells. Cells were treated with MG132 for 8 h before collection. Then Sox2 was immunoprecipitated with anti-Sox2 antibody and immunoblotted with anti-HA antibody. NC represents the empty vector control. **h** Immunoblotting of Sox2, KLF4 and E2F1 in MEFs with CUL4A, CUL4B knockout or WT control. Quantification of protein levels relative to vinculin is shown. **i** MEFs were treated with CHX (10 μg/ml), and collected at the indicated times for western blot. Quantification of Sox2 levels relative to vinculin is shown. **$P < 0.01$, two-way ANOVA test. The representative images are shown from three independent experiments. Unprocessed original scans of blots are shown in Supplementary Fig. 9

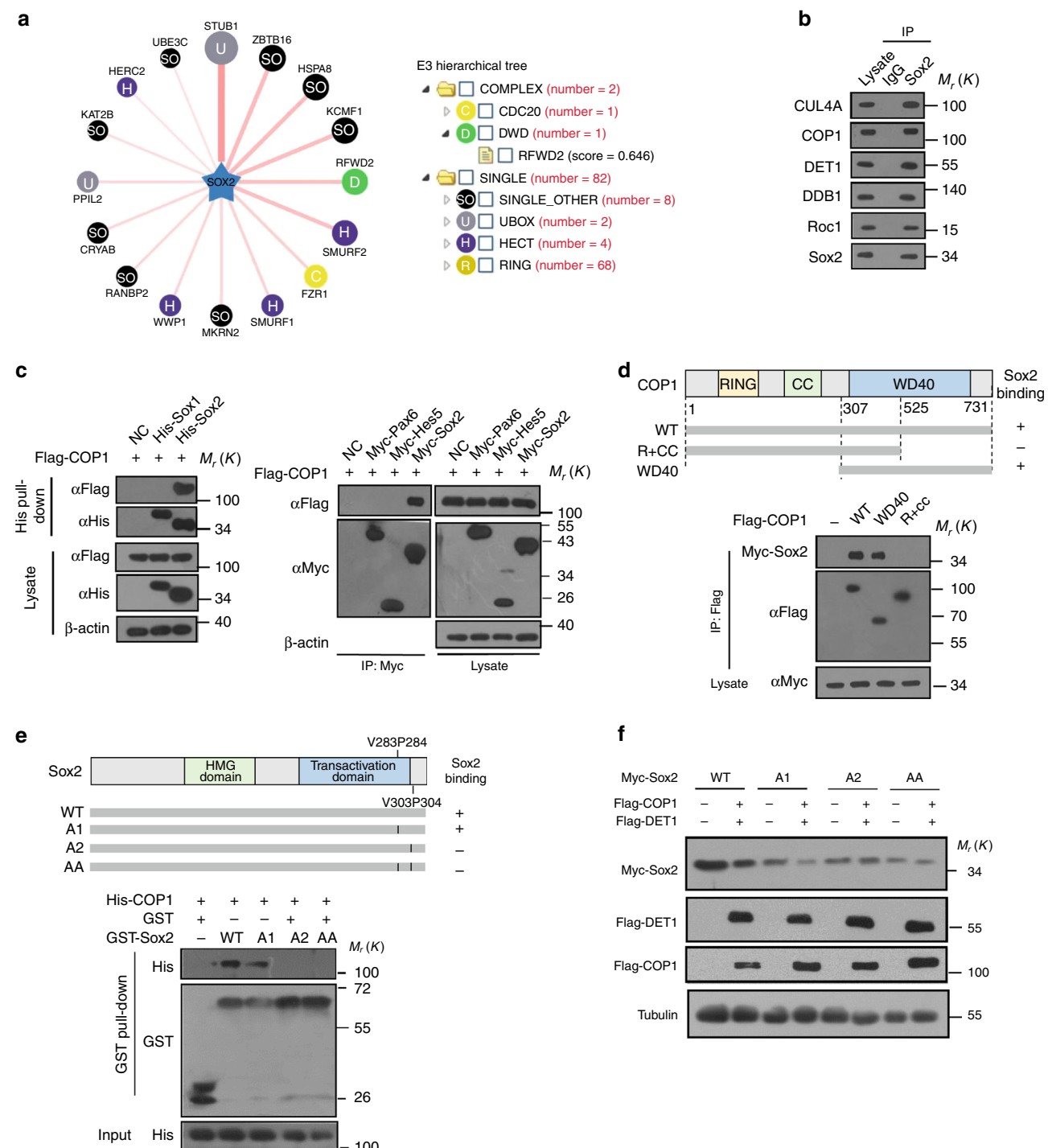

transcription factor in cell reprogramming, was not affected (Fig. 1h). The half-life of Sox2 was prolonged in CUL4A-deficient MEFs, compared to wild type (WT) cells (Fig. 1i). Taken together, these results suggested that CUL4A contributes to Sox2 ubiquitylation in NPCs.

**COP1 interacts with Sox2 directly**. CUL4A has been documented to act as a scaffolding protein for modular CRL4 complexes which promote the ubiquitylation of a variety of substrates[21,22]. The C-terminal end of CUL4A binds to RING finger protein ROC1/RBX1 to recruit ubiquitin-bearing E2 enzymes[22]. At its N-terminal end, CUL4A bind to DDB1. DDB1

functions as a substrate adaptor protein by forming a connection between the CUL4 scaffold and multiple substrate receptors known as DCAFs. DCAFs, also called WD40 domain containing proteins (DWD or CDW proteins), provide one determinant for substrate recognition[21,22]. To identify the DCAF recruiting Sox2 to the CUL4A core complex, we employed UbiBrowser[25], an integrated bioinformatics platform to predict the proteome-wide human E3-substrate network based on naïve Bayesian network (http://ubibrowser.ncpsb.org). We queried Sox2 as substrate, and the 84 predicted E3 ligases in UbiBrowser were presented in Fig. 2a and Supplementary Data 1. These E3s were classified into different families and shown in the E3 family hierarchical tree

**Fig. 2** CUL4A^DET1-COP1 interacts with Sox2 and regulates its stability. **a** Network view of E3–Sox2 interactions (left panel) and the E3 hierarchical tree for Sox2 (right panel). UbiBrowser was employed to explore the E3 ligases for Sox2. The representative predicted E3 ligases surround Sox2. The node colors and characters reflect the E3 type. The edge width, the node size, and the edge shade are corrected with the confidence score. The predicted E3s and their position in the E3 family hierarchical tree was presented. In this tree, texts in each circle (just like "U", "D" and "SO") represent the E3 family. The number in the bracket following each E3 family represents the number of corresponding predicted E3–Sox2 interaction. **b** NPCs cell lysates were subjected to immunoprecipitation with control IgG or anti-Sox2 antibodies and detected CUL4A, COP1, DET1, DDB1, Roc1, and Sox2 protein levels. **c** The lysates of HEK293T cells transfected with indicated constructs were subjected to immunoprecipitation with anti-Myc or Histidine tag-specific affinity resin (agarose beads). The immunoprecipitates or the eluates were then blotted. **d** Overview of the structures of COP1 wild type and different truncates. HEK293T cells were co-transfected with Myc-Sox2 and the indicated COP1 truncates. The lysates were collected and subjected to immunoprecipitation with anti-Flag. The immunoprecipitates were then blotted. **e** Overview of the structure of Sox2 wild type and different VP mutants. Recombinant proteins (His-COP1, GST-Sox2, GST-Sox2-A1, GST-Sox2-A2, and GST-Sox2-AA) were expressed and purified. GST-Sox2 bound to glutathione-Sepharose 4B beads was incubated with His- COP1 for 24 h at 4 °C. Then the beads were washed and proteins were eluted, followed by western blotting. **f** HEK293T cells were transfected with indicated constructs. The lysates were collected and blotted with anti-Flag and anti-Myc antibody. The representative images are shown from three independent experiments. Unprocessed original scans of blots are shown in Supplementary Fig. 9

(Fig. 2a, right panel). Among the 84 predicted E3s, COP1 was the unique DCAF protein, and given a relative high Likelihood ratio (LR): 3.98. Therefore we thought that COP1 might be the potential regulator of Sox2. By further investigating the supporting evidence, we found that Sox2 contains two VP motifs which match the noncanonical VP degron and COP1 always recognizes the substrate's degron using the WD40 domain[26,27]. COP1 had been proved to directly interact with ETV1 or c-Jun and promote ubiquitylation by assembling a multisubunit ubiquitin ligase containing CUL4A, DDB1, and DET1 (CUL4A^DET1-COP1)[27]. We proposed that the CUL4A^DET1-COP1 complex is responsible for Sox2 ubiquitylation and degradation.

To validate the speculation on the interaction between Sox2 and CUL4A^DET1-COP1, we designed a variety of biochemical experiments. We found that CUL4A, COP1, DET1, DDB1, and ROC1 were co-immunoprecipitated (Co-IP) with Sox2 in NPCs (Fig. 2b). Ectopic COP1 specifically interacted with Sox2, but not with Pax6, Sox1, and Hes5, key transcription factors in NPCs (Fig. 2c). COP1 also did not interact with the stem cell transcription factors Nanog and Oct4 (Supplementary Fig. 2a). GST pull-down assay was performed to confirm the direct interaction between COP1 and Sox2 in cell-free system (Supplementary Fig. 2b).

Then, we mapped the Sox2-binding region of COP1. Deletion analysis demonstrated that the WD40 repeats of COP1 mediated the physical interaction with Sox2 (Fig. 2d). Furthermore, we determined the COP1-binding domain of Sox2. Sox2 contains two potential noncanonical degrons (COP1-binding VP motifs, V283/P284, and V303/P304) in its C-terminal region (Supplementary Fig. 2c). We mutated the conserved residues in the VP motif and generated three mutants V283A/P284A (here named A1), V303A/P304A (here named A2), and V283A/P284A/ V303A/P304A (here named AA) (Fig. 2e). With Co-IP assays, we found that combined V283A/P284A and V303A/P304A mutation (Sox2 AA) abolished the interaction of Sox2 with COP1, but had no significant effect on the interaction of Sox2 with WWP2 (Supplementary Fig. 2d). Further, we analyzed the contribution for the binding of each VP motif independently with GST pull-down assays. The results showed that the mutant V283A/P284A retained the binding ability of Sox2 to COP1, while the V303/P304 mutant and the combined mutant AA lost the ability to bind to COP1 (Fig. 2e). Consistently, ectopic COP1 expression downregulated the levels of wide type Sox2 and the A1 mutant but not those of Sox2 A2 and AA mutants (Fig. 2f). Therefore, we conclude that COP1 recognizes Sox2 mainly via the V303/P304 motif.

Of note, COP1 knockdown by siRNA blocked the interaction between Sox2 and CUL4A, DET1, ROC1, or DDB1 (Fig. 3a), while CUL4A or DET1 knockdown did not affect the binding of

COP1 and Sox2 in cells (Fig. 3b). Additionally, we observed that Sox2 was up-regulated in cells with depletion of COP1, DET1, CUL4A, DDB1, or ROC1 (Supplementary Fig. 3a). Co-expression of COP1 and DET1 remarkably decreased Sox2 protein level compared with COP1 expression alone (Fig. 3c). Furthermore, in the presence of DET1, ectopic expression of COP1 resulted in Sox2 down-regulation in a dose-dependent manner (Fig. 3d). COP1 and DET1 did not affect the levels of Pax6, Sox1, and Hes5 (Supplementary Fig. 3b). All of the results suggested that CUL4A^DET1-COP1 acts as the E3 ligase complex for Sox2 degradation and COP1 directly binds to Sox2 and plays a role as a specific substrate receptor (Fig. 3e).

**CUL4A^DET1-COP1 ubiquitylates Sox2 for degradation**. We wonder if the COP1 and DET1 affect protein stability by ubiquitylating Sox2. In Fig. 4a, b, we observed that COP1 or DET1 knockdown prolonged half-life of Sox2 and decreased the ubiquitylation of Sox2 in cells. In vitro ubiquitylation assay showed that purified CUL4A, COP1, and DET1 proteins ubiquitylated Sox2 in the cell-free system (Fig. 4c). To identify which lysine residues are subjected to ubiquitylation, we performed a series of ubiquitylation assays using Sox2 KR mutants. Human Sox2 protein contains 17 lysine residues (K10, K35, K42, K58, K65, K73, K80, K87, K95, K103, K109, K115, K117, K121, K122, K124, and K245). Initially, we generated a Sox2 K0 mutant in which all the 17 lysines were mutated to arginines. Then each mutated arginine was mutated back to lysine to generate 17 mutants containing one single lysine. These mutants of Sox2 were tested for ubiquitylation assays by COP1. As shown in Supplementary Fig. 4a, no ubiquitylated Sox2 was detected in cells transfected with Sox2 K0 mutant. Interestingly, we also observed that Sox2 was ubiquitylated by COP1 on multiple lysines including K58, K65, K73, K80, K87, K95, K103, K109, K122, K124, and K245.

To evaluate the contribution of WWP2 and COP1 for Sox2 ubiquitylation in NPCs, we generated NPCs with depletion of WWP2, COP1, or combined both of them. Compared with negative control, the half-life of Sox2 was prolonged in NPCs with COP1 depletion alone or combined depletion of COP1 and WWP2 (Supplementary Fig. 4b). While in WWP2 depletion cells, the half-life of Sox2 was unchanged compared with the control cells (Supplementary Fig. 4b). Ubiquitylation assay showed that Sox2 ubiquitylation was blocked by knockdown of COP1 and combined COP1 with WWP2, while in WWP2-depleted cells, Sox2 ubiquitylation remained unchanged (Supplementary Fig. 4c). These results suggested that in NPCs, COP1 but not WWP2 promotes Sox2 ubiquitylation and degradation.

The methylation of mouse Sox2 on K119 by Set7 had been reported to be required for WWP2-mediated ubiquitylation[13]. So we wondered if Set7-mediated methylation is essential for

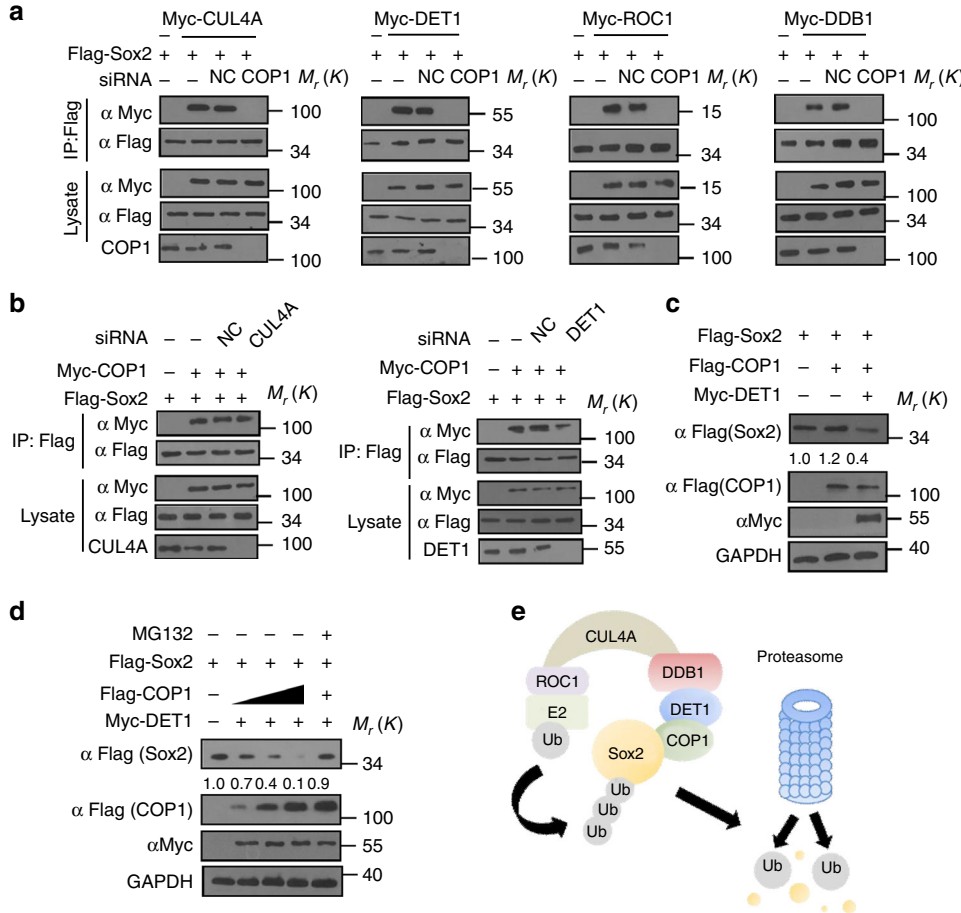

**Fig. 3** COP1 interacts with Sox2 directly. **a** Cells were transfected with indicated constructs and siRNA. Flag-Sox2 was immunoprecipitated with anti-Flag and immunoblotted with anti-Myc. **b** HEK293T cells were transfected with indicated constructs and siRNA, and Sox2 was immunoprecipitated with anti-Flag and immunoblotted with anti-Myc. **c** HEK293T cells were transfected with indicated constructs and western blot was performed to measure the expression of Flag-Sox2, Flag-COP1, and Myc-DET1. **d** Increasing amounts of Flag-COP1 were co-transfected together with Flag-Sox2 and DET1 into HEK293T cells and ectopic Flag-Sox2 expression was detected. **e** The predicted work model of CUL4A$^{\text{DET1-COP1}}$ for Sox2 degradation. The representative images are shown from three independent experiments. Unprocessed original scans of blots are shown in Supplementary Fig. 9

CUL4A$^{\text{DET1-COP1}}$-mediated Sox2 ubiquitylation. As shown in Fig. 4d, the mutation of K119R blunted Sox2 decrease induced by ectopic WWP2, but not for COP1/DET1. Consistently, siRNA targeting Set7 reversed the decrease of Sox2 level mediated by WWP2 overexpression. However, in COP1/DET1 or CUL4A overexpressed cells, Set7 knockdown had no such effect on Sox2 level (Fig. 4e). Thus, Sox2 ubiquitylation mediated by CUL4A-$^{\text{DET1-COP1}}$ is independent of Sox2 K119 methylation by Set7.

Further, we performed luciferase reporter assay to determine the influence of CUL4A$^{\text{DET1-COP1}}$-mediated ubiquitylation on the transcriptional activity of Sox2. As shown in Supplementary Fig. 4d and e, we observed that co-expression of COP1 and DET1 inhibited the transcriptional activation of Sox2 WT on FGF4 promoter-driven luciferase, but had no significant effect on Sox2-AA mutant.

**CUL4A$^{\text{DET1-COP1}}$ promotes NPCs differentiation**. Subsequently, we further investigated the functional roles of CUL4A$^{\text{DET1-COP1}}$ in NPCs differentiation. With the cell differentiation, the expression of CUL4A and COP1 increased gradually that inversely correlated with Sox2 (Fig. 5a). COP1 silencing by short hairpin RNA (shRNA) in NPCs increased Sox2 level and inhibited cell differentiation marked by decreased TUJ1 and increased Nestin level (Fig. 5b, c and Supplementary Fig. 5a).

Further, we performed the rescue assay to confirm the specific regulation of COP1 on Sox2 upon NPCs differentiation. Firstly, we knock down Sox2 in NPCs with depletion of COP1, and observed that combined knockdown of Sox2 with COP1 significantly blunted the decrease of TUJ1 induced by COP1 knockdown alone (Fig. 5d and Supplementary Fig. 5b). Then we simultaneously overexpressed COP1 and Sox2-AA in NPCs, and found that ectopic Sox2-AA partially rescued COP1-induced NPCs differentiation (Fig. 5e and Supplementary Fig. 5c). Taken together, these data showed that the E3 complex CUL4A$^{\text{DET1-COP1}}$ is specific for Sox2 ubiquitylation and promotes human NPCs differentiation.

**OTUD7B deubiquitylates and stabilizes Sox2**. DUBs maintain protein stability by specifically deconjugating ubiquitin from targeted proteins. The human genome encodes at least 98 DUBs, which can be grouped into six families: UCHs, USPs, OTUs, Josephins, JAMMs, and MCPIP, reflecting the need for specificity in their function [28]. Altered DUB function has been related to several diseases, including neurodegeneration disease and cancer [28]. So far, the DUBs involved in the regulation of Sox2 ubiquitylation and stability remain undefined. So we attempted to identify the potential DUBs that remove ubiquitin chains from Sox2. To this end, a subset of the ovarian tumor (OTU) family of DUBs were employed to screen. Notably, ectopic expression of

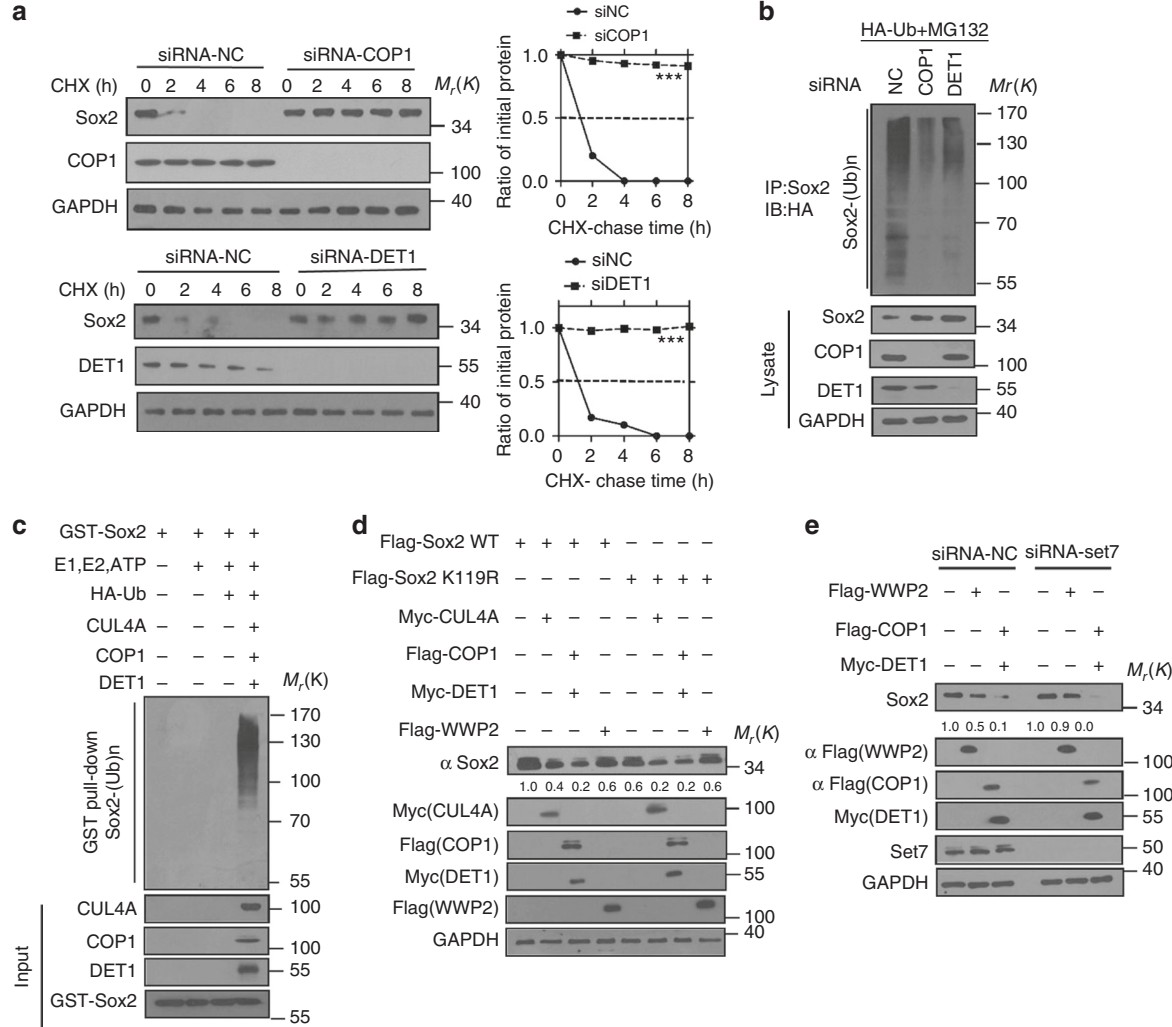

**Fig. 4** CUL4A$^{DET1-COP1}$ ubiquitylates SOX2. **a** HEK293T cell line with knockdown of COP1 (upper panel) or DET1 (lower panel) were treated with CHX (10 μg/ml), and collected at the indicated times for western blot. Quantification of Sox2 level relative to tubulin is shown. Results are shown as mean ± s.d. $n =$ 3 independent experiments. **$P < 0.01$, two-way ANOVA test. **b** HA-Ub was co-transfected with indicated siRNA into HEK293T cells. Cells were treated with MG132 for 8 h before collection. Sox2 was immunoprecipitated with anti-Sox2 and the ubiquitylated Sox2 proteins were immunoblotted with anti-HA. **c** Cell-free Sox2 ubiquitylation assay. The purified GST-Sox2, CUL4A, COP1, and DET1 proteins were incubated with commercial E1, UbE2D3 (E2), and ubiquitin for 2 h at 37 °C. The mixtures were subjected to GST pull-down and western blot with anti-His antibody. **d** Immunoblotting of Flag-Sox2 in cells transfected with indicated constructs. Quantification of Sox2 level is shown under the blot. **e** Cells were transfected with indicated siRNA and the cell lysates were subjected to western blot to detect Sox2 expression. Quantification of relative protein level is shown under the blot. The representative images are shown from three independent experiments. Unprocessed original scans of blots are shown in Supplementary Fig. 9

OTUD7B, but not other DUB family members examined, specifically upregulated Sox2 protein level (Fig. 6a). OTUD7B (also called Cezanne-1) is a typical member of the subfamily of OTU DUBs[29,30] and has been documented to regulate T-cell activation and tumor development by NF-κB, mTOR, or hypoxia signaling[31–36]. However, the functional roles of OTUD7B in stem cells is unknown. We examined the effect of OTUD7B on Sox2 stability. The results showed that increased ectopic expression of OTUD7B, but not OTUD7A, elevated Sox2 levels in a dose-dependent manner (Supplementary Fig. 6a). The half-life of Sox2 was shortened in cells with OTUD7B silence, but prolonged in cells with OTUD7B overexpression, while depletion or overexpression of OTUD7A showed minimal effects on Sox2 level (Fig. 6b, Supplementary Fig. 6b and c). Unlike Sox2, the levels of Nanog, Oct4, Sox1, Pax6, and Hes5 were not influenced by OTUD7B (Supplementary Fig. 6d), indicating the specificity.

We went on to examine the influence of OTUD7B on Sox2 ubiquitylation. ShRNA targeting OTUD7B but not OTUD7A

increased Sox2 ubiquitylation (Fig. 6c). Also, we found that OTUD7B-mediated Sox2 deubiquitylation required the DUB enzymatic activity of OTUD7B, as a catalytically inactive mutant H358R, but not C194S, failed to remove the ubiquitin linkage from Sox2 (Fig. 6d and Supplementary Fig. 6e). Although both H358 and C194 have been suggested essential for hydrolysis of K11-linked or K48-linked polyUb[30], C194S mutant was not observed to affect OTUD7B catalytic activity in our system. It might be due to the potential plasticity of OTUD7B results in dramatic conformational transitions along the catalytic cycle[28] and the formatiom of Cezapo, the autoinhibited state of OTUD7B by the conformation of the Cys-loop2 (residues 187–193)[30].

Then we detected the interaction between Sox2 and OTUD7B at endogenous (Fig. 6e) and exogenous (Supplementary Fig. 6f) levels and mapped the Sox2-binding region of OTUD7B to the C-terminal zinc finger (ZF) domain (Fig. 6f). To further confirm the direct binding of Sox2 to OTUD7B ZF domain, we

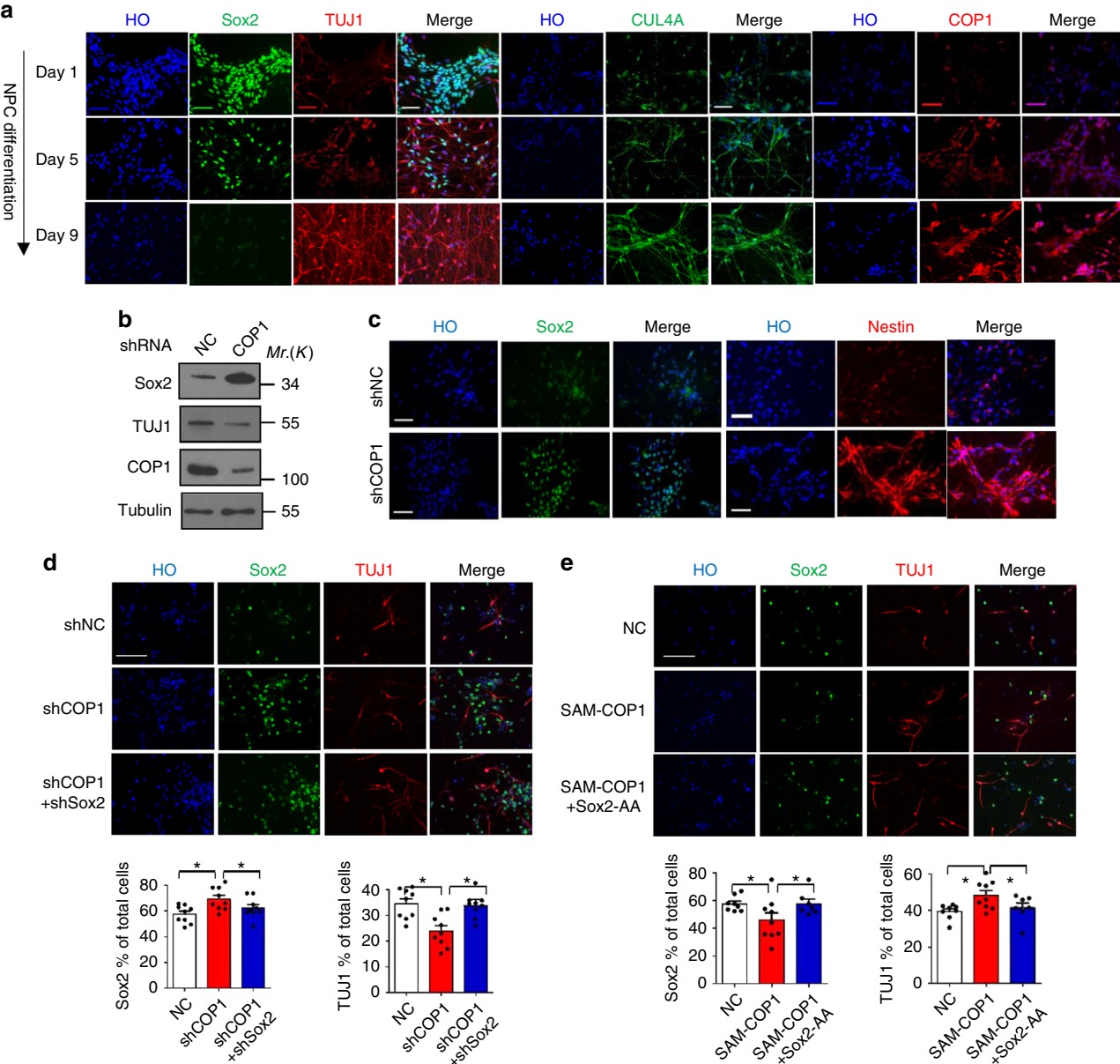

**Fig. 5** COP1 promotes NPCs differentiation in a Sox2-dependent manner. **a** NPCs were plated in Matrigel-coated six-well plate for neuronal differentiation. For the indicated times, fixed and stained with anti-Sox2 (green), anti-COP1 (red), anti-TUJ1 (red), or CUL4A (green)-specific antibodies. Nuclei were counterstained with Hoechst 33342 (HO, blue). Scale bar, 50 μm. **b**, **c** NPCs were infected with lentiviruses expressing COP1 shRNA or negative control (NC) shRNA for 126 h. Immunoblotting (**b**) and immunostaining (**c**) were performed for Sox2, COP1, TUJ1, or Nestin. Scale bar, 50 μm. **d** NPCs with double knockdown of COP1 and Sox2 were generated by lentivirus infection. Immunostaining were performed for Sox2, TUJ1. Scale bar, 50 μm. The ratio of TUJ1-positive or Sox2-positive cells were quantified. Results are shown as mean ± s.d. Each error bar shows the standard deviation of numbers of positive cells in 10 fields of view. *$P < 0.05$, Student's $t$-test. **e** NPCs with co-expression of COP1 and Sox2-AA were generated by lentivirus infection. Cas9-based activator of COP1 (SAM-COP1) were used to activate COP1 expression in NPCs. Immunostaining were performed for Sox2 and TUJ1. Scale bar, 50 μm. The ratio of TUJ1-positive or Sox2-positive cells were quantified. Results are shown as mean ± s.d. Each error bar shows the standard deviation of numbers of positive cells in 10 fields of view. *$P < 0.05$, Student's $t$-test. The representative images are shown from three independent experiments. Unprocessed original scans of blots are shown in Supplementary Fig. 9

purified recombinant GST-tagged OTUD7B truncates, including GST-OTUD7B-N (1-359), GST-OTUD7B-C (360–843), GST-OTUD7B-△ZF (1–810), and GST-OTUD7B-ZF (811–843) (Fig. 6g). The result of GST pull-down assay indicated that Sox2 interacted with OTUD7B ZF domain directly in cell-free system (Fig. 6g).

Furthermore, we examined the effect of OTUD7B-mediated deubiquitylation on Sox2 transcriptional activity on FGF4 promotor-driven luciferase. As shown in Supplementary Fig. 6g,

we observed that the expression of OTUD7B but not OTUD7A enhanced the transcriptional activation by Flag-Sox2 from a FGF4 promoter-driven luciferase reporter. OTUD7B H358R mutant, but not C194S, failed to enhance Sox2 transcriptional activity (Supplementary Fig. 6h). These results demonstrated that OTUD7B specifically stabilize Sox2 by deubiquitylation in cells. Subsequently, using OTUD7B-deficient MEFs, we confirmed that OTUD7B deficiency reduced Sox2 level (Fig. 7a), shortened the half-life of Sox2 protein (Fig. 7b) and elevated Sox2 ubiquitylation

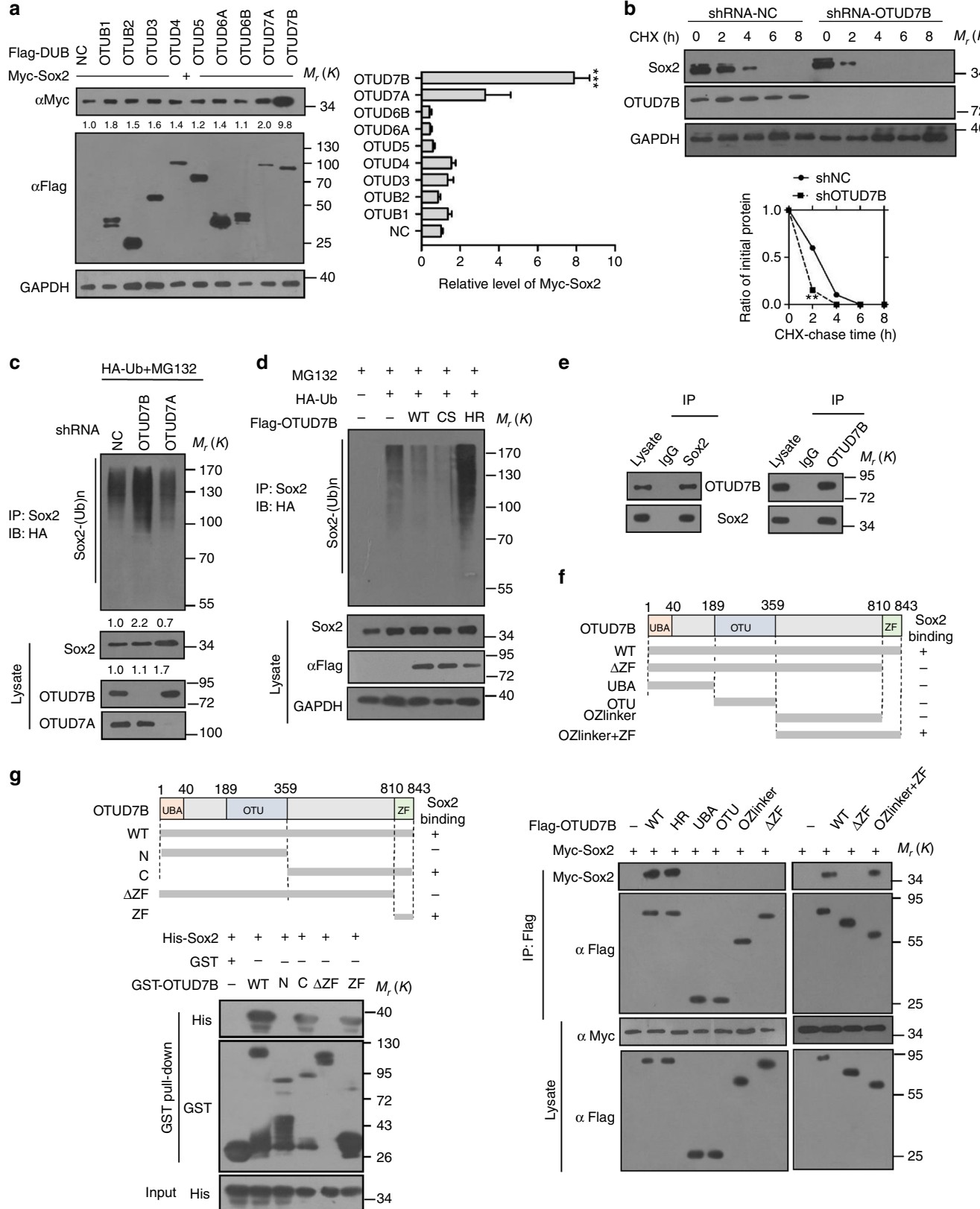

(Fig. 7c). Thus, OTUD7B is a bona fide DUB that removes ubiquitin from Sox2.

**OTUD7B maintains NPCs stemness in a Sox2-dependent manner.** In order to investigate the function of OTUD7B in

NPCs, we firstly examined the dynamic expression of OTUD7B during NPCs differentiation. OTUD7B level decreased gradually that consistently correlated with Sox2 with the cell differentiation (Fig. 7d). Then we generated NPCs with stable knockdown of OTUD7B (Supplementary Fig. 7a). NPCs properties were

**Fig. 6** OTUD7B deubiquitylates and stabilizes Sox2. **a** The indicated OTU subfamily DUBs were each transfected into HEK293T cells. 48 h later, cell lysates were subjected to western blot. Quantification of relative Sox2 levels is shown. Results are shown as mean ± s.d. Each error bar shows the standard deviation of the value from three independent experiment. ***$P < 0.001$ compared with NC group, Student's $t$-test. **b** Half-life analysis of Sox2in cells with OTUD7B knockdown. Cells transfected with indicated shRNA were treated with 10 μg/ml CHX, and collected at the indicated times for western blot. Quantification of Sox2 levels relative to tubulin is shown. Results are shown as mean ± s.d. $n = 3$ independent experiments. **$P < 0.01$, two-way ANOVA test. **c** HEK293 cells transfected with the indicated shRNA were treated with MG132 for 8 h before collection. Sox2 was immunoprecipitated with anti-Sox2 and immunoblotted with anti-HA. **d** Cells transfected with the indicated constructs were treated with MG132 for 8 h before collection. The whole-cell lysate was subjected to immunoprecipate with anti-Sox2 and western blot with anti-HA antibody. **e** The lysates of NPCs were subjected to immunoprecipitation with IgG, anti-Sox2, or anti-OTUD7B antibody and immunoblotted. **f** Overview of OTUD7B structures (upper panel) and the interaction between Sox2 and OTUD7B truncations (lower panel). Cells transfected with the indicated constructs were subjected to immunoprecipitation with anti-Flag antibodies. The lysates and immunoprecipitates were then blotted. **g** GST pull-down was performed to confirm the binding of OTUD7B with Sox2 in cell-free system. The purified His-Sox2 proteins were incubated with indicated purified OTUD7B truncations. The representative images are shown from three independent experiments. The mixtures were subjected to GST pull-down and western blot. Unprocessed original scans of blots are shown in Supplementary Fig. 9

determined by immunofluorescence (IF) and immunoblot. OTUD7B silencing in NPCs decreased Sox2 level and promoted cell differentiation (Fig. 7e, f).

Subsequently, we wondered if the function of OTUD7B is dependent on its regulation on Sox2 modification in NPCs. We expressed ectopic Sox2-AA in NPCs with depletion of OTUD7B, and observed that Sox2-AA blunted OTUD7B knockdown-induced NPC differentiation (Fig. 7g and Supplementary Fig. 7b). Then we overexpressed OTUD7B in NPCs with depletion of Sox2 and found that Sox2 silencing partially rescued the decrease of NPCs differentiation resulted from OTUD7B overexpression (Fig. 7h and Supplementary Fig. 7c). So, OTUD7B takes a role as the DUB specific for Sox2 stabilization and NPCs maintenance.

**COP1 and OTUD7B coordinate to control NPCs differentiation**. We attempted to determine how CUL4A$^{DET1-COP1}$ controls Sox2 stability together with OTUD7B during NPCs differentiation. Western blotting (Fig. 8a) and IF (Fig. 8b and Supplementary Fig. 8a) assays showed that both OTUD7B and Sox2 levels gradually declined during NPC differentiation, while CUL4A and COP1 increased, suggesting that both decreased OTUD7B and increased COP1 contribute to Sox2 ubiquitylation and degradation during neuronal differentiation. Simultaneous targeting of OTUD7B and COP1 resulted in intermediate levels of Sox2 ubiquitylation (Fig. 8c). Overexpression of OTUD7B together with COP1 and DET1 abolished the effect of ectopic COP1-DET1 on Sox2 level (Supplementary Fig. 8b), indicating that OTUD7B antagonizes COP1-DET1 in regulating Sox2.

Subsequently, we tested the effect of modifications of Sox2 level on NPCs differentiation by chemical perturbations of the UB/DUB system. Given that the specific inhibitor targeting CUL4A$^{DET1-COP1}$ or OTUD7B is unavailable, we used unspecific DUB inhibitors PR-619[37], b-AP-15[38], and the inhibitor of NEDD8-activating enzyme (NAE) MLN4924 which had been demonstrated to suppress neddylation of Cullins, and inactivate Cullin-RING ligases[39]. The results showed that compared with control, the treatment of MLN4924 promoted NPC maintenance marked by increased Sox2 level and decreased neuN level, while PR-619 and b-AP-15 enhanced cell differentiation marked by decreased Sox2 level and increased neuN level (Supplementary Fig. 8c).

Previous studies showed that during ESCs differentiation, Set7 methylates Sox2 at K119, which results in WWP2-mediated Sox2 ubiquitylation and degradation[13]. Additionally, Ube2S mediates polyubiquitylation at the Sox2-K123 residue and regulates ES cells self-renewal and differentiation toward the neural ectodermal lineage in mice[17]. In Supplementary Fig. 2c and Supplementary Fig. 8d and e, we examined the half-life of Sox2 and the protein level of related regulatory enzymes, including WWP2, Set7, Ube2S, COP1, DET1, CUL4A, and OTUD7B in ESCs, NPCs, and

neurons, respectively. We observed that during the differentiation from ESCs to NPCs, the level of Set7 and Ube2S increased, while that of WWP2 decreased significantly. During the differentiation from NPCs to neurons, the expression levels of COP1, DET1, and CUL4A increased and that of OTUD7B decreased, suggesting that diverse regulatory machines take roles in different stage of stem cell differentiation.

Taken together, we revealed an unrecognized dynamic mechanism regulating Sox2 protein level during NPCs differentiation. We demonstrated that CUL4A$^{DET1-COP1}$ promotes Sox2 degradation through ubiquitylation and induces differentiation of NPCs, while OTUD7B stabilizes Sox2 through deubiquitylation and promotes maintenance of NPCs (Fig. 8d). Our results indicate that OTUD7B and CUL4A$^{DET1-COP1}$ exert opposite roles in regulating Sox2 protein stability at the post-translational level.

## Discussion

The neurogenesis in embryo and adult are precisely controlled and governed by multiple interdependent extracellular factors and cell intrinsic molecular pathways, among which ubiquitylation has emerged as a potent regulatory principle that determines protein turnover[40]. Aberrant ubiquitin signaling contributes to a variety of brain disorders like X-linked mental retardation, schizophrenia, or Parkinson's disease[40]. HECT-type E3 ligases SMAD ubiquitination regulatory factor 1 (Smurf1)[41] and HUWE1 (HECT, UBA, and WWE domain containing 1)[42] have emerged as important regulators of axon acquisition and morphology. Recent studies have highlighted that polyubiquitin modification of key molecular is an essential pathway to regulate NPCs properties. Huang et al.[43] reported that β-TrCP and HAUSP-mediated repressor-element-1-silencing transcription factor (REST) ubiquitylation governs the maintenance and differentiation of NPCs. F-box and WD repeat domain containing-7 (Fbw7), the substrate recognition component of an SCF (complex of SKP1, CUL1, and F-box protein)-type E3 ligase has been proved to control neural stem cell differentiation and progenitor apoptosis via Notch and c-Jun[44]. In addition, deubiquitylating enzymes USP22 and USP27x modulate Hairy and enhancer of split 1 (Hes1) protein dynamics by removing ubiquitin molecules, and thereby regulate neuronal differentiation of stem cells[45].

Our study indicated that RING-type E3 ligase complex CUL4A$^{DET1-COP1}$ promotes NPCs differentiation towards neuron by ubiquitylating Sox2, the core transcriptional factor of cell reprogramming. COP1, as the substrate receptor interacts directly with and ubiquitylates Sox2. While WWP2, the known E3 ligase for negative regulating Sox2 in ESCs[13] was difficult to detect in NPCs, indicating WWP2 might not play a major role in this process. Notably, we also revealed that CUL4A$^{DET1-COP1}$-mediated Sox2 ubiquitylation seems to be independent of Sox2

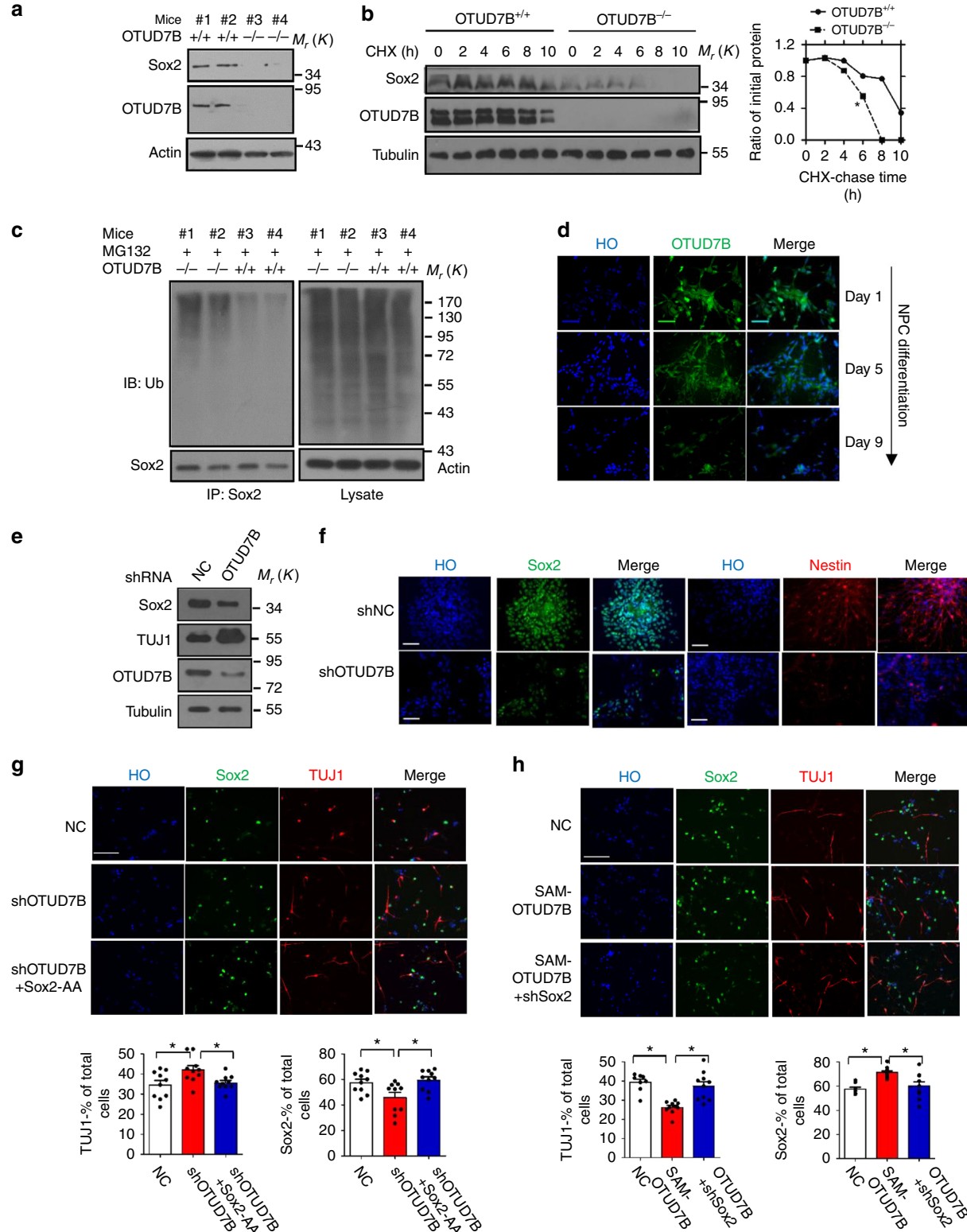

methylation by Set7. In this regard, CUL4A[DET1-COP1] and WWP2 function through different manners. Besides that, a most recent study has revealed that ubiquitin-conjugating enzyme Ube2S is involved in the regulation of Sox2 protein stability in ESCs[17]. However, Ube2S had been observed to reinforce the self-renewing and pluripotent state of ES cells. Importantly, it also represses Sox2-mediated ES cell differentiation toward the neural

ectodermal lineage. Thus, Ube2S regulates ESCs properted independent on Sox2[17]. So, we propose that during different physiological processes, Sox2 stability and activity were fine-tuned through different molecular machinery (Fig. 8d).

Mammalian COP1 is a member of COP-DET-FUS protein family and it mainly resides in the nucleus, although a small amount may also be present in the cytosol[46]. COP1 protein

**Fig. 7** OTUD7B maintains NPCs stemness. **a** Immunoblotting of Sox2 in MEF with OTUD7B knockout and WT control. **b** The MEF cells were treated with CHX (10 μg/ml), and collected at the indicated times for western blot. Quantification of Sox2 levels relative to tubulin is shown. Results are shown as mean ± s.d. $n = 3$ independent experiments. **$P < 0.01$, two-way ANOVA test. **c** Sox2 ubiquitylation in MEF with OTUD7B knockout and WT control. MEFs were treated with MG132 for 8 h and the lysates were subjected to immunoprecipitation with anti-Sox2 and immunoblotted with anti-Ub. **d** NPCs were plated in Matrigel-coated six-well plate for neuronal differentiation. For the indicated times, fixed and stained with anti-OTUD7B (green). Nuclei were counterstained with HO (blue). **e**, **f** NPCs were infected with lentiviruses expressing OTUD7B shRNA or negative control (NC) shRNA for 126 h. Immunoblotting (**e**) and immunostaining (**f**) were used to detect the level of Sox2, TUJ1, or Nestin in NPCs. Scale bar, 50 μm. **g** NPCs with stable OTUD7B knockdown and Sox2-AA overexpression were generated with lentivirus infection. Immunostaining were performed for Sox2, TUJ1. Scale bar, 50 μm. The ratio of TUJ1-positive or Sox2-positive cells were quantified. Results are shown as mean ± s.d. Each error bar shows the standard deviation of numbers of positive cells in 10 fields of view. *$P < 0.05$. Student's $t$-test. **h** NPCs with stable OTUD7B overexpression and Sox2 knockdown were generated with lentivirus infection. Cas9-based activator of OTUD7B (SAM-OTUD7B) were used to activate OTUD7B expression in NPCs. Immunostaining were performed for Sox2, TUJ1. Scale bar, 50 μm. The ratio of TUJ1-positive or Sox2-positive cells were quantified. Results are shown as mean ± s.d. Each error bar shows the standard deviation of numbers of positive cells in 10 fields of view. *$P < 0.05$. Student's $t$-test. The representative images are shown from three independent experiments. Unprocessed original scans of blots are shown in Supplementary Fig. 9

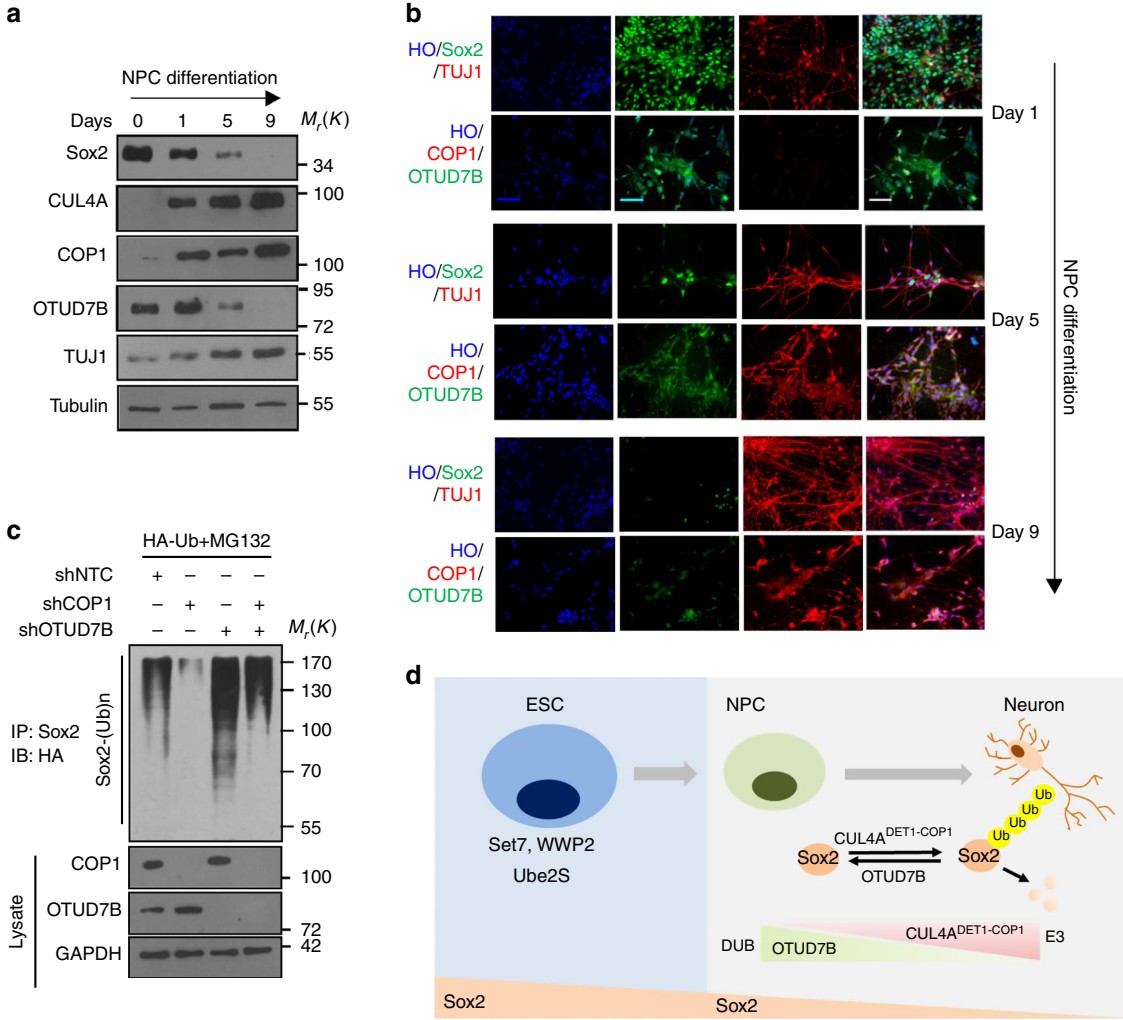

**Fig. 8** COP1 and OTUD7B coordinate to control NPCs differentiation. **a**, **b** NPCs were plated in Matrigel-coated six-well plate for neuronal differentiation. For the indicated times, immunoblotting (**a**) and immunofluorescence (**b**) of Sox2, CUL4A, COP1, OTUD7B and TUJ1 were performed. **c** Cells transfected with HA-Ub and indicated shRNA were treated by MG132 for 8 h, and the lysates were immunoprecated with anti-Sox2 antibody and immunoblotted with anti-HA. **d** Summary of our findings. OTUD7B stabilizes Sox2 through deubiquitylation and maintains the stemness of NPCs, while CUL4A^DET1-COP1 promotes Sox2 degradation through ubiquitylation and induces differentiation of NPCs. OTUD7B and CUL4A^DET1-COP1 exert opposite roles in regulating Sox2 protein stability at the post-translational level. The representative images are shown from three independent experiments. Unprocessed original scans of blots are shown in Supplementary Fig. 9

contains a RING domain, a coiled-coil domain and seven WD40 repeats and the main cellular functions of COP1 are probably mediated through ubiquitylation and degradation of its substrates[46]. Most of the identified substrates of COP1 are transcriptional factors, including c-Jun[47], ETS variants (ETV1/4/5)[26], and FOXO1 protein[48]. Since both c-Jun and ETV are known as oncoproteins (especially in prostate cancer), COP1 has been suggested to act as a potential tumor suppressor gene. In mice, the complete loss of COP1 function results in embryonic lethality, whereas studies of a partial loss of COP1 function have provided in vivo evidence of its tumor suppressor role[46,47,49]. Here we show that another transcriptional factor, Sox2, is a new substrate of COP1 in NPCs and COP1 regulates NPC differentiation through Sox2, if not all, at least partially. So far as we know, this study provides the first evidence to unravel a role of COP1 in stem cell regulation. Moreover, the possible role of COP1–Sox2 interaction in the development of human cancers is worthy of investigations in the future.

Importantly, we revealed that DUBs OTUD7B utilizing its C-terminus ZF domain interacts directly with Sox2 and stabilizes Sox2 protein through deubiquitylation for the first time. Although recent studies identified several substrates of OTUD7B, including TRAF3 and GβL (a component of mTOR complex)[31,32], and the role of OTUD7B in regulating immune response and metabolic stress response accordingly, the intact cellular function of OTUD7B still remains mysterious. The current findings showed that OTUD7B, but not other OTU family members we examined, upregulated the expression of Sox2. Also, OTUD7B did not stabilize Pax6, Hes5, Oct4, nor Nanog. Thus, OTUD7B seems to be a specific DUB for Sox2 and maintains NPCs property through the removal of ubiquitin from Sox2. Interestingly, OTUD7B was highly expressed in both ESCs and NPCs, suggesting that it might play a wide role in regulation of both stages, which needs further deep studies.

With the recent advances in the technology of stem cell therapy, ESCs, NPCs, and induced neural stem cells have been used to treat neurodegenerative diseases[50,51]. As regenerative resources, the use of NPCs requires sufficient proliferation and appropriate differentiation. Here we reveal an unrecognized regulatory mechanism on the ubiquitylation of Sox2, the key transcriptional factor controlling NPCs maintenance and differentiation. The utility of NPCs or of derived differentiated cell types would be expanded if we govern Sox2 level by perturbations of the UB/DUB system. We tested the effect of modifications of Sox2 level on NPCs differentiation by chemical perturbations of the UB/DUB system. MLN4924 as the inhibitor of NAE promoted NPC maintenance, while PR-619 and b-AP-15 enhanced cell differentiation. These data suggest that the strategy targeting UB/DUB system might have potential benefit in the treatment of neurodegenerative diseases.

Collectively, this study has uncovered a crucial post-translational control mechanism to regulate the key stem cell transcriptional factor Sox2 for cell fate determination of NPCs which will pave the way to improving the treatment of neural degenerative diseases.

## Methods

**Cell culture of human neuron progenitor cells**. Human NPCs (WA09, passages 20–24; WA01, passages 30–35) were maintained under E8 medium (Gibco, A1516401)) by coating with vitronectin (Gibco, A14700). For 3–4 days of maintainence, cells were dissociated by dispase (Gibco, 17105041) to form embryoid bodies (EBs), and then cultured in NIM (500 ml of NIM contains 5 ml of N2 supplement, 5 ml of NEAA, and 490 ml of DMEM/F12). After floating for 7 days, EBs were attached on vitronectin-coated surfaces. Rosette structures can be observed at day 10–16. At day 16, rosette clones were detached by a 1 ml pipette manually. Non-neuroepithelial clones can be removed at this stage. Neurospheres were continuous floated in NIM, and then dissociated by TrypLE

(Gibco, 12604021) and plated on Matrigel coated six-well plate for neuronal differentiation. Samples were collected before attachment and after 1 day, 5 days, and 9 days attachment.

**Culture of MEFs**. We obtained the CUL4A/B knock-out MEFs from Dr. Pengbo Zhou (Weill Cornell Medical College). OTUD7B knock-out MEFs were purchased from Model Animal Research Center of Nanjing University, China. We added the relevant description in the Methods section. MEFs were cultured in complete DMEM at 37 °C in a 5% $CO_2$ incubator. Cell yield and viability were determined with trypan blue staining.

**Plasmid and constructs**. The human DUB plasmids and HA-Ub were generated as described previously[52]. The mutants of Sox2 K119R was gifted from Prof. Jiemin Weng (East China Normal University). The vectors containing open reading frames of Sox2, DET1, Nanog, or Oct4 were purchased from OriGene and Pax6, Hes5 and Sox1 were purchased from Vigene Biosciences. For Co-IP assay and cell transfection, full-length Sox2, Sox1, Pax6, Hes5, Oct4, Nanog, COP1, DET1, CUL1, CUL2, CUL3, CUL4A, CUL4B, CUL5, OTUD7A, OTUD7B and the truncates examined were cloned into pCMV-Myc vector, pCMV-Flag vector, or pcDNA3.1-Myc-His (−) C vector as indicated. For GST pull-down assay, Sox2, COP1, OTUD7B and the truncates were cloned into pGEX-6p-1 vectors or pET-28a (+). The mutants including Myc-Sox2-A1, A2, AA, and OTUD7B-C194S, H358R were generated using KOD-Plus-Mutagenesis Kit (TOYOBO) according to the manufacturer's instructions. Sox2-K0 mutant in which all the 17 lysines were mutated to arginines was generated by Sangon Biotech (Shanghai). Then each mutant arginine was mutated back to lysine to generate 17 mutants containing one single lysine (K10, K35, K42, K58, K65, K73, K80, K87, K95, K103, K109, K115, K117, K121, K122, K124, or K245 only) using KOD-Plus-Mutagenesis Kit (TOYOBO) according to the manufacturer's instructions. All the vectors constructed above were PCR-amplified with Q5 Hot Start High-Fidelity 2 × Master Mix (NEB). The primers for constructs were presented in Supplementary Data 2.

**RNA interference**. Control siRNA and the on-target individual siRNAs were synthesized by Shanghai GenePharm. Each target gene employed two effective sequences as below: siWWP2-#1 (5′-AAGGUGCAUAAUCGUCAACCUCG-3′), siWWP2-#2 (5′-AAACUGCUUUGGUTGGAAGAUCCC-3′); siCOP1-#1 (5′-GCUGGAGUUACAAAGAAGATT-3′), siCOP1-#2 (5′-CCACCAUCAAUGTAACUCCAT-3′); siCUL1-#1 (5′-AAGUCCAGAUAUAUGGGAGCUCTA-3′), siCUL1-#2 (5′-AAUAAACAGGUAACAAAUGCUGT-3′); siCUL2-#1 (5′-AAUACGUCGAAAGAGCAACAUGG-3′), siCUL2-#2 (5′- AAAUGAUCGUGGUGGAGAAGACC-3′); siCUL3-#1 (5′-GGCCUUUCCGATUGACCAUGGATG-3′), siCUL3-#2 (5′-GGGAUCAUCUACGGCAAACUCTA-3′); siCUL4A-#1 (5′-AAGAGACUAAUUGCUUAUAUGCT-3′), siCUL4A-#2 (5′-AAGCUGGAAGGCAUGUUCA AGGA-3′); si-CUL4B-#1 (5′-AAGAAGAGAAAGUAAACAGCAG-3′), si-CUL4B-#2 (5′-AAAGGAAUGGUGAAGCAAUUGAT-3′); si-CUL5-#1 (5′-GAGCA CTACGTTATTTAGAAA-3′), siCUL5-#2 (5′-GGTATGCCAGCGGATTA TGTA-3′). Non-tageting siRNAs (5′-UUCUCCGAACGUGUCACGU-3′). The OTUD7A and OTUD7B shRNA clones were purchased from OriGene. shOTUD7A-#1 (5′-TTCTACATGATCCTATGACTCTT-3′), shOTUD7A-#2 (5′-CCACTGTGTGCACGAGCTGTAAA-3′); shOTUD7B-#1 (5′-GCAGCAGACACAGCAGAAT-3′), shOTUD7B-#2 (5′-TGGGTATACACAGAAGATGA-3′). Non-target control: 5′-TTCTCCGAACGTGTCACGT-3′. Transfections were performed with Lipofectamine2000 (Invitrogen), and the knockdown efficiency was verified by quantitative PCR or Western blot.

**Lentivirus packaging**. Lentiviral knockdown vectors (pLKO.1 puro) carrying shRNA targeting human COP1, WWP2, OTUD7B, OTUD7A, or Sox2 were constructed, which all employed two effective sequences as follows: shCOP1-#1 (Sense, 5′-GCTGGAGTTACAAAGAAGATT-3′), shCOP1-#2 (Sense, 5′-CCACC ATCAATGTAACTCCAT-3′); shWWP2-#1 (Sense, 5′-GGTGCATAATCGTCAA CCTCG-3′), shWWP2-#2 (Sense, 5′-ACTGCTTTGGTTGGAAGATCCC-3′); shSox2-#1 (Sense, 5′'-CTGCCGAGAATCCATGTATAT-3′), shSox2-#2 (Sense, 5′-CCATGGGTTCGGTGGTCAA-3′). The effective sequences of shOTUD7A, shOTUD7B and non-target control are listed as above. Lentiviral over-expressed vector (pCDH-MCS-T2A-Puro-MSCV) carrying Sox2-AA were constructed. Cas9-based activators of COP1 or OTUD7B were inserted to lenti sgRNA (small guide RNA) puro backbone (#73795, Addgene) to generate the lentiviral vectors (lenti-SAM-COP1 and lenti-SAM-OTUD7B) according to the SAM target sgRNA cloning protocol. The lentiviruses were packaged in 293FT cells and were used to infect cells in the presence of Polybrene. 5 days after infection, NPCs were collected to detect by Western blot or IF staining.

**Antibodies**. Antibodies were used for immunoblot, immunoprecitication (IP) and IF staining. All antibodies were purchased as follows. Anti-Sox2 antibodies for IB (#3579, 1:500 dilution; scsc-365823, 1:300 dilution) were purchased from Cell Signaling Technology or Santa Cruz Biotechnology. Anti-Sox2 for IF (MAB2018, 1:1000) was purchased from R&D Systems. Anti-Myc (M047-3, 1:2000) and anti-HA (M180-3, 1:1000) were purchased from MBL. Anti-Flag (F1804, 1:1000), anti-CUL1 (C1771, 1:500), anti-CUL2 (SAB4200207), anti-CUL3 (C0871), anti-CUL4A

(C0371, 1:1000 for IB, 1:1000 for IF), anti-CUL4B (C9995, 1:1000), anti-CUL5 (SAB2701084, 1:5000), anti-β III-Tubulin (TUJ1) (T58660, 1:500 for IB, 1:2000 for IF) were purchased from Sigma. Anti-Sox2 (sc-20088, 1:300 for IB), anti-COP1 (sc-166799, 1:50 for IB, 1:50 for IF), anti-Ub (sc-8017, 1:1000), anti-GST (sc-374171, 1:1000), anti-His (sc-8036, 1:300), anti-Nestin (sc-21247, 1:1000 for IF) were purchased from Santa Cruz. Anti-OTUD7B (16605-1-AP, 1:2000 for IB) were purchased from Proteintech. Anti-OTUD7B (NBP1-88095, 1:50 for IF) was purchased from Novus Biologicals. Anti-DET1 (A9974, 1:1000), anti-Set7 (A9985, 1:300), anti-Ube2S (A4685, 1:1000) were purchased from ABclonal Technology. Anti-WWP2 (ab103527, 1:500) was purchased from Abcam. Anti-α-Tubulin (TA-10, 1:10,000), anti-GAPDH (TA-08, 1:1000), anti-β-actin (TA-09, 1:2000) were purchased from Zsbio. Hoechst 33342 (H1399, 1:4000) and all the fluorescence-conjugated secondary antibodies were purchased from Invitrogen.

**Immunoprecipitation and immunoblotting**. Cells were lysed with HEPES lysis buffer (20 mM HEPES, pH7.2, 50 mM NaCl, 0.5% Triton X-100, 1 mM NaF and 1 mM dithiothreitol) supplemented with protease-inhibitor cocktail (Roche). The protein lysates were incubated with the indicated primary antibodies and protein A/G agarose (Santa Cruz, sc-2003) at 4 °C. The immunocomplexes were then washed with HEPES lysis buffer three times. Then immunoblot was carried out with primary antibodies to identify the target protein.

**Immunofluorescent staining**. NPCs cultured on the BD Matrigel-coated cover glasses were induced by NIM for differentiation or infected with lentiviruses, fixed in 4% paraformaldehyde for 15 min, permeablized, blocked by blocking buffer containing 0.1% donkey serum, and 0.3% Triton X-100 for 30 min, incubated with the primary antibodies overnight at 4 °C, and incubated with the fluorescence-conjugated secondary antibodies at room temperature for 1 h. Nuclei were counterstained with Hoechst 33342.

**In vivo ubiquitylation assay**. Human NPCs NIM-induced differentiated cells or HEK293T cells transfected by indicated constructs are subjected to in vivo ubiquitylation assays. Cells were treated with the proteasome inhibitor MG132 (20 μM; Sigma) for 8 h and then were collected in lysis buffer (50 mM Tris–HCl, pH7.4, 150 mM NaCl, 5 mM EDTA, 1% NP-40 supplemented with protease inhibitors cocktail) and lysates were pre-cleared by centrifugation at 15,000×g for 15 min. Total protein lysate (500 μg) was then incubated with indicated antibody for 3 h and protein A/G-agarose beads for a further 8 h at 4 °C. After three washes, ubiquitinated Sox2 was detected by immunoblotting with anti-HA (or anti-Ub) monoclonal antibody.

**In vitro ubiquitylation assay**. His-COP1, His-DET1, His-CUL4A, and GST-Sox2 were expressed in HEK293T and purified. Indicated proteins were pretreated at 30 °C for 30 min. Afterwards, 0.7 μg of E1, 0.9 μg of UbcH5c, 12 μg of HA-ubiquitin, 0.7 μg of His-COP1, His-DET1, His-CUL4A, and 1.6 μg GST-Sox2 are resolved in ubiquitylation assay buffer. The reactions were stopped by the addition of SDS–PAGE sample buffer. The reaction products were resolved by SDS–PAGE gel and probed with the indicated antibodies.

**GST pull-down assay**. Recombinant proteins with GST or His tag were expressed in BL21 E. coli. GST-tagged protein bound to glutathione-Sepharose 4B beads (GE healthcare) was incubated with His-tagged protein for 24 h at 4 °C. Then the beads were washed with GST-binding buffer (100 mM NaCl, 10 mM Tris, 50 mM NaF, 2 mM EDTA, 0.5 mM Na₃VO₄, and 1% Nonidet P40) four times and proteins were eluted, followed by western blotting.

**Dual luciferase reporter assay**. HEK293 cells were seeded in 24-well plates at a density of $1 \times 10^5$. After 24 h, the cells were transfected using Lipofectamine 2000 (Invitrogen). The promoter region of FGF4 targeted by Sox2 was cloned into the pGL3-Luc vector (Promega, Madison, WI, USA), and subsequently co-transfected with Myc-Sox2 and Flag-OTUD7B or Flag-COP1, DET1 and CUL4A, together with the pRL-SV40 Renilla luciferase construct. Single overexpression of Sox2 was used as controls. Cell extracts were prepared 48 h after transfection and the luciferase activity was measured by using the Dual Luciferase System (Promega, Madison, WI, USA) and a Centro LB960 96-well luminometer (Berthold Technologies, Bad Wildbad, Germany).

The promoter region of FGF4 targeted by Sox2 is as below:
5′AGACCGTCTTTTAGAAAATAACAAGAAGAAAAGACATT
TCAACTGTCTTCTCCCCAACACTCTTGGAGCCTAGGGCCTGGATTT
AAAAAACACAAAATCTTATTGTCCTGTGAGCCACCAGACAGAAAGG
AAGTTTGGGAGGAGCTCCCACCTCAGCTTTGGGGTGTGGCTTCTTCC
ATCTTGGGCTGTGGTACAGAATAGTATTTTAAGTATCCCATTAGCATCC
AAACAAAGAGTTTTCTAAAGGAATGTGAAAGACAAAAAAAAAAA
AATGCCAATGAGATTTTCCAGTCTTGCTGTCTGTAGCCTCCCATAAAGTT
AATTCGGGAGGTTGCTCAGAAGTCTCCCAGCAGAGTCTTA-3′

**Protein degradation analysis**. For Sox2 half-life assay, human NPCs NIM-induced differentiated cells or HEK293T cells transfected by indicated constructs or

siRNA. 24 h later, cells were treated with the protein synthesis inhibitor cycloheximide (Sigma, 10 μg/ml) for the indicated durations before harvest. Western blot was carried out to detect Sox2 protein level in cells.

**Real-time quantitative PCR**. Total RNA samples were isolated and then subjected for reverse transcription and Q-PCR analysis in the 7500HT Fast Real-Time PCR System (AB Applied Biosystems) using the following primer pairs: Sox2: sense: 5′-TACAGCATGTCCTACTCGCAG-3′, antisense: GAGGAAGAGGTAACCACA GGG; WWP2: sense: 5′-CGCAACTATGAGCAGTGGCA-3′, antisense: GGTCGTGCGAGTGTTATGGT-3′.

**Query for interactions of E3–Sox2 in UbiBrowser**. UbiBrowser (http://ubibrowser.ncpsb.org) is designed to allow users to explore the predicted and reported E3–substrate interactions for a query E3 or substrate, and ubiquitylation sites for substrate protein. We queried Sox2 as substrate, and the 84 predicted E3 ligases in UbiBrowser were presented in Supplementary Data 1.

**Statistical analysis**. All results are shown as the mean ± s.d. of multiple independent experiments. Detailed n values for each panel in the figures are stated in the corresponding legends. A student's t-test, or Kruskal–Wallis one-way or two-way ANOVA tests were used for statistical analyses. All statistical analyses were performed with GraphPad Prism 5 software. All statistical tests were two-sided, and P values < 0.05 were considered to be statistically significant.

## Data availability
The authors declare that all the relevant data supporting the findings of this study are available within the article and its Supplementary Information files, or from the corresponding author on reasonable request.

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

## Acknowledgements

We thank Dr. Jiemin Weng (East China Normal University, Shanghai, China) and Pengbo Zhou (Weill Cornell Medical College) for kindly providing materials. This work was jointly supported by Beijing Science and Technology Major Project (Z181100004118004), the Strategic Priority Research Program of the Chinese Academy of Sciences (XDA16010306 and XDB29020000), and Chinese National Natural Science Foundation Projects (31330021 and 81521064).

## Author contributions

The project was conceived by L.Z. The experiments were designed by L.Z., Y.L., C.P.C., and F.H. Most of the experiments were performed by C.P.C., Y.Z., C.W., and F.Y. NPCs culture and analysis were contributed by F.Y., C.W., and Y. Z. The protein ubiquitylation assays, the protein interaction assays were performe by Y.Z., C.P.C., and H.L., Q-PCR assays were performed by C.L. The establishment of knockout mice and isolation and culture of MEF were performed by Y.C., H.L., and C.W. The data were analyzed by Y.Y., C.H.L., C.P.C., Y.Z., C.W., W.W., H.L., F.H., Y.L., and L.Z. The manuscript was written by L.Z., C.P.C., and Y.L.

## Additional information

**Competing interests:** The authors declare no competing interests.

