## [Peer Review File · Nature Communications]

Reviewers' Comments:

Reviewer #1:

Remarks to the Author:

CUL4A DET1-COP1 and OTUD7B Regulate Sox2 Proteostasis and Govern Neural Progenitor Cell Differentiation

Tight regulation of key transcription factors is critical to regulate the proliferation and differentiation of stem cells. Sox2 plays prominent role in the maintenance of ESCs and NPCs. In contrast to Sox2 proteostasis in ESCs, those in NPCs remained elusive. From this point of view, Cui et al have explored underlying mechanisms of Sox2 ubiquitination in NPCs. The authors have found that the CUL4A DET1-COP1 complex mediates the ubiquitination of Sox2 in NPCs, whereas OTUD7B promote the de-ubiquitination of Sox2, and the tightly regulated Sox2 levels by the ubiquitination plays critical roles in the maintenance of NPCs. The findings are important and the manuscript is well-written. However, there are several major concerns, and the authors need to address those points before the publication in Nature communications.

(Major Points)

1, Fig1, the selectivity of CUL4A DET1-COP1

The authors claimed that the CUL4A DET1-COP1 selectively regulates the ubiquitination of Sox2 in NPCs. However, this point is not clear enough. In Fig1 H, the knockout of CUL4A increased the levels of E2F1. Moreover, they did not address the interaction between the CUL4A DET1-COP1 and other key transcription factors in NPCs, such as Pax6, Sox1, and Hes1/5. Although they addressed the interaction between CUL4A DET1-COP1 and Nanog or Oct4, these transcription factors are not expressed in NPCs, therefore in the context of Sox2-proteostasis in NPCs, they are not adequate controls.

2, Fig2. Direct interaction between Sox2 and COP1.

A series of Co-IP experiments with knockdown approaches nicely showed that COP1 serves as a linker. Based on these data, the authors claimed that COP1 directly binds to Sox2. However, it is not clear if this interaction is direct or indirect. They can address its direct interaction using their cell free system.

3, Fig3. Fig5. Functional assay for CUL4A DET1-COP1/ OTUD7B

It is important to assess if the main target of CUL4A DET1-COP1-dependent proteostasis is Sox2 in functional assay. Fig1g,h. Knockdown of COP1 showed the increased levels of Sox2 and Nestin. It would be nice to test if the knockdown of COP1 also inhibits the differentiation of NPCs and knockdown of Sox2 can reverse the phenotype. In the same line, test if exogenous expression of COP1/CUL4A can induce the differentiation of NPCs and further addition of Sox2-AA can rescue the phenotype. The same approached will be applicable for OTUD7B. Those experiments will further support their conclusion that the levels of Sox2 is selectively regulates in NPC by ubiquitination.

(Minor pints)

1. Fig1d, the rationale to check DDB1 was not explained.

2, Most of WB signals show clear difference between control and manipulation, but some of them are not clear enough if the results support the authors conclusion. It would be a good idea to quantify the signal intensities of WB in Fig1h, Fig2g, Fig3d, Fig4b,

3, It may be interesting to compare the temporal dynamics of Sox2 ubiquitination/degradation between NPCs and ESCs, to explain/discuss why those cells need to use the different pathway to control the levels of Sox2, how those dynamics could contribute the maintenance of ESCs and NPCs, respectively.

4. Please quantify all blots and calculate differences using a relevant statistical test and report the

p-values. This should be done for all comparisons not limited to the ones mentioned below, but with special attention to the ones below:

- Fig. 1c, please quantify, calculate the correlation coefficient and p-value (corrected for multi-testing across all of the culins) for each comparison.
- Fig. 4a, it is difficult to assess the increase in Sox2, please quantify and add p-value for the test.
- Fig. 4c, it is difficult to assess the differences between the pattern with an sh for OTUD7A and OTUD7B, please quantify and report the p-value.

5. In fig 1d, knockdown of the adaptor DDB1 also increased Sox2 levels, please discuss the implications within the relevant results section.

6. There are numerous grammatical errors throughout the manuscript.

Reviewer #2:

Remarks to the Author:

Cui et al performed a study on the regulation of the SOX2 protein during the differentiation of neural progenitor cells by ubiquitylation pathways leading to proteosomal degradation. SOX2 is a key factor maintaining a progenitor state in many cell lineages and elucidating the multi-layered mechanisms regulating its activity is of general interest. The authors identify the ubiquitin ligases and de-ubiquitination enzymes involved in this process. They further perform a variety of biochemical and cell biological assays to establish CUL4A, DET1/COP1 and OTUD7B as main components controlling Sox2 ubiquitination and stability. Overall the study contains a large amount of work (with a focus on co-immunoprecipitations to map interactions and Ub conjugations and western blots/immunocytochemistry to monitor protein levels), which is mostly well executed. However, the manuscript is difficult to read for the lack of subheadings and structure. It is certainly useful to pinpoint the exact E3 ligase (COP1) and DUB (OTUD7B) enzymes as well as the conjugation sites and interaction domains mediating the interactions with Sox2 but such insights are mainly of relevance to researchers with a focus on the regulation of protein degradation pathways. To a more general audience it comes as no surprise that the Sox2 turnover is post-translationally regulated but the relevance to stem cell biology and neurogenesis is not immediately clear. The authors close their manuscript by stating that their insights could have a 'significant impact on improving the treatment of neural degenerative diseases.' To justify this claim it could be shown that modifications of Sox2 levels by chemical perturbations of the UB/DUB system could expand the utility of NPCs or of derived differentiated cell types. In its present form the manuscript appears to be better suited for a more general journal.

Specific comments:

- 1) Acronyms are at times confusing and used interchangeably which can be confusing.
- 2) Line 84: Description for "Supplementary Fig. 1b" does not match to the figure.
- 3) Line 137: Marking mistake - "(Fig. 3d,e)" should be "(Fig. 2d,e)".
- 4) Line 141: Marking mistake- "(Fig. 3e)" should be "(Fig. 2e)".
- 5) Number of biological replicates should be indicated and the statistical tests used should be mentioned.
- 6) Differences in band intensities are sometimes not obvious and should be supported by quantifications. For example, in Fig.3e, the difference in Sox2 levels between "siRNA-NT" and "siRNA-Set7" does not appear to be significant
- 7) Line 214-216: "Then we detected the interaction between Sox2 and OTUD7B at endogenous (Fig. 4e) and exogenous (Supplementary Fig. 4f) levels and mapped the Sox2-binding region of OTUD7B to the ZF (zinc finger) domain (Fig. 4f)." The evidence for the identification of Sox2-binding region of OTUD7B is not convincing. Since the ZF is a DNA/RNA domain, interactions could simply be mediated by DNA/RNA scaffolds rather than being direct. Interactions with purified

domains could further strengthen this point.

8) The methods lack detail. For example, the sequences of the FGF4 promoter used for the luciferase assay are unclear as is the amount of plasmids used in the assays.

9) Abbreviations need to be expanded at least once, suppliers of reagents and plasmid IDs should be given.

Reviewer #3:

Remarks to the Author:

This is a nice manuscript describing the control of Sox2 protein stability during NPCs differentiation. Through high-quality biochemical experiments, the authors conclude that the CRL4-DET1-COP1 ligase complex is responsible for timely Sox2 protein degradation. Additionally, the authors identify OTUD7B as the de-ubiquitinase for Sox2. Interestingly, the CUI4 complex together with COP1 increases in expression during NPCs differentiation, while OTUD7B decreases, and these factors positively correlates with NPCs differentiation and negatively correlates with Sox2 protein level. Deregulation of Sox2 protein level by shRNA targeting COP1 or OTUD7B impairs NPCs differentiation. Therefore, the manuscript provides an important post-translational regulatory step for NPCs differentiation in a CRL4-DET1-COP1- and OTUD7B-dependent manner.

Major comments:

- Page 6. Lines 125-127. The references reported by the authors (#24,#25) identify a canonical COP1's degron that is '[D/E]-X(2,3)-V-P-[D/E]' and not just 'V-P' as reported by the authors for Sox2. Hence, the degron that they identify is not a canonical degron and their statement is incorrect. As such, the mapping for the degron can be considered preliminary. I would suggest to remove these data unless supported by more extensive evidence (e.g., mapping the interaction of Sox2 N-term and C-term truncation mutant with COP1; analyze of the contribution for the binding of V283/P284 and V303/P304 independently, and provide additional protein stability analysis of the mutants eventually identified).

- Page 8. Lines 163-164. Fig. 3d shows Sox2 protein levels, not its half-life. Therefore, it is incorrect to state that Sox2 K119R is subject to degradation by COP1/DET1. The same holds true for Fig. 3e. Moreover, the authors already shown that WWP2 was not expressed in their experimental condition. Nevertheless, it would be important to evaluate the contribution of the ligase WWP2 in Sox2 degradation and compare it to CRL4-DET1-COP1-mediated degradation. Given that the authors show RNA expression of WWP2 in supplementary Fig. 1a, I suggest to test the half-life of Sox2 comparing different conditions to evaluate the contribution of each ligase complex to the stability of Sox2 protein:

- 1- after depletion of WWP2
- 2- after depletion of COP1 (and/or DET1)
- 3- after combined depletion of WWP2 and COP1 (or DET1)

Minor Comments:

- Page 4. 'Suppl. Fig. 1b', it seems that the authors are referring to 'Fig. 1c'.

- Fig. 2d. The order of the mutant and their labeling differ from the ordering and labeling in Fig. 2e. It would be better to change the order of the mutants in Fig. 2d (as the order of the same mutants in Fig. 2e) and to label Fig. 2e with the same labels used in Fig. 2d.

- Fig. 2h. The figure is mislabeled (the labeling is shifted down and covered by the first panel), hence, of difficult interpretation/evaluation.

- Page 7. Line 141. 'Fig. 3e', it seems that the authors are referring to 'Fig.2e'.

- Page 8. Line 176. '...COP1 decreased gradually', it seems that the authors meant: '...COP1 increased gradually'.

- Fig. 3f. The figure shows the juxtaposition of 3 channels. The readings of each separate channel should be reported at least in supplementary data.

- Fig 4i. Why there are 4 lanes in Fig. 4i and only the last 2 lanes show a difference? Moreover, the levels of Sox2 are not shown.

- Fig 4l. Labeling is not properly aligned.

- Fig. 5b. The figure is shows the juxtaposition of 3 channels. The readings of each separate channel should be reported at least in supplementary data.

RESPONSE TO REVIEWERS' COMMENTS:

Reviewer #1 (Remarks to the Author):

Tight regulation of key transcription factors is critical to regulate the proliferation and differentiation of stem cells. Sox2 plays prominent role in the maintenance of ESCs and NPCs. In contrast to Sox2 proteostasis in ESCs, those in NPCs remained elusive. From this point of view, Cui et al have explored underlying mechanisms of Sox2 ubiquitination in NPCs. The authors have found that the CUL4A DET1-COP1 complex mediates the ubiquitination of Sox2 in NPCs, whereas OTUD7B promotes the de-ubiquitination of Sox2, and the tightly regulated Sox2 levels by the ubiquitination plays critical roles in the maintenance of NPCs. The findings are important and the manuscript is well-written. However, there are several major concerns, and the authors need to address those points before the publication in Nature communications.

Response: We thank the reviewer for the kind comments on the thoroughness of our manuscript and for recognizing the novelty and significance of this study. According to the concerns, we have revised the manuscript and the responses are listed below point by point.

(Major Points)

1. Fig 1, the selectivity of CUL4A^{DET1-COP1}

The authors claimed that the CUL4A^{DET1-COP1} selectively regulates the ubiquitination of Sox2 in NPCs. However, this point is not clear enough. In Fig1 H, the knockout of CUL4A increased the levels of E2F1. Moreover, they did not address the interaction between the CUL4A^{DET1-COP1} and other key transcription factors in NPCs, such as Pax6, Sox1, and Hes1/5. Although they addressed the interaction between CUL4A^{DET1-COP1} and Nanog or Oct4, these transcription factors are not expressed in NPCs, therefore in the context of Sox2-proteostasis in NPCs, they are not adequate controls.

Response: We sincerely thank the reviewer for such insightful comments. That's important points. The cell cycle regulatory transcription factor E2F1 has been reported to be targeted for ubiquitination and degradation by CUL4A^{Cdt2}, another E3 ligase complex [Zielke N et al, 2011]. So it is reasonable that Cul4A deficiency resulted in an increase of the protein level of E2F1. In this study, we examined E2F1 level to confirm the effect of Cul4A knockout in MEF cells and this is a positive control. In the revised manuscript, we added the description "knockout of CUL4A but not CUL4B remarkably increased the level of Sox2 as well as that of E2F1 which has been shown as a direct substrate of CUL4A^{Cdt2} ligase".

In the revised manuscript, we examined the interaction of COP1 and Pax6, Sox1 and Hes5, key transcription factors in NPCs with Co-IP assays, and the results showed that COP1 could not bind to Pax6, Sox1 or Hes5 in cells (revised Fig. 2c). Additionally, we also examined the effect of COP1 or OTUD7B overexpression on the level of Pax6, Sox1 and Hes5. The results showed that COP1 and OTUD7B had no significant effects on the levels of Pax6, Sox1 and Hes5 (Supplementary Fig. 3b and Supplementary Fig. 6d).

2. Fig 2. Direct interaction between Sox2 and COP1.

A series of Co-IP experiments with knockdown approaches nicely showed that COP1 serves as a linker. Based on these data, the authors claimed that COP1 directly binds to Sox2. However, it is not clear if this interaction is direct or indirect. They can address its direct interaction using their cell free system.

Response: We thank the reviewer for bringing up this constructive suggestion. In the revised manuscript, we used GST pull-down assay to detect the interaction of COP1 and Sox2. As shown in revised Supplementary Fig. 2b, COP1 binds to wild type Sox2 in cell free system. Therefore, this interaction is direct.

3. Fig 3. Fig 5. Functional assay for CUL4A^{DET1-COP1}/ OTUD7B

It is important to assess if the main target of CUL4A^{DET1-COP1}-dependent proteostasis is Sox2 in functional assay. Fig 1g, h. Knockdown of COP1 showed the increased levels of Sox2 and Nestin. It would be nice to test if the knockdown of COP1 also inhibits the differentiation of NPCs and knockdown of Sox2 can reverse the phenotype. In the same line, test if exogenous expression of COP1/CUL4A can induce the differentiation of NPCs and further addition of Sox2-AA can rescue the phenotype. The same approached will be applicable for OTUD7B. Those experiments will further support their conclusion that the levels of Sox2 is selectively regulates in NPC by ubiquitination.

Response: We thank the reviewer for raising this outstanding concern. We carried out a series of rescue assays to determine if Sox2 is the main target of CUL4A^{DET1-COP1}-dependent proteostasis in NPCs. Using COP1 shRNA, we demonstrated that COP1 knockdown reduces TUJ1 level in NPCs, indicating that COP1 depletion inhibits the differentiation of NPCs (revised Fig. 5b, c and Supplementary Fig. 5a). Then we knockdown Sox2 in NPCs with COP1 depletion, and overexpressed Sox2-AA in NPCs with COP1 overexpression, and examined the level of Sox2 and TUJ1 through immunofluorescence analysis. Combined knockdown of Sox2 with COP1 significantly blunted the decrease of TUJ1 induced by COP1 knockdown alone (revised Fig. 5d and Supplementary Fig. 5b). While simultaneous expression of ectopic Sox2-AA and COP1 reversed the increase of TUJ1 level induced by COP1 overexpression alone (revised Fig. 5e and Supplementary Fig. 5c), indicating that

COP1 induces NPCs differentiation largely dependent of Sox2.

At the same time, we expressed ectopic Sox2-AA in NPCs with OTUD7B knockdown, and silenced Sox2 in NPCs with OTUD7B overexpression. Exogenous Sox2-AA expression suppressed TUJ1 increase induced by OTUD7B silencing (revised Fig. 7g and Supplementary Fig. 7b), while Sox2 knockdown blocked the effect of exogenous OTUD7B on cell differentiation in NPCs (revised Fig. 7h and Supplementary Fig. 7c), suggesting that OTUD7B plays a role in NPCs differentiation also dependent of Sox2.

(Minor pints)

1. Fig. 1d, the rationale to check DDB1 was not explained.

Response: We thank the reviewer for this important concern. As shown in Fig. 1c, the protein level of CUL4A, but not that of other cullins, was upregulated during NPC differentiation and inversely correlated with Sox2, suggesting that CUL4A Cullin-RING Ligase (CRL) complex might be involved in NPCs differentiation and Sox2 degradation. To further confirm the effect of CUL4A E3 complex in NPCs, we examine Sox2 level in cells with depletion of CUL1, CUL2, CUL3, CUL4A or DDB1. DDB1 (damaged DNA binding protein 1) has been documented as an integral component of CUL4A/4B complex to link CUL4A/4B and DCAF and regulates a set of vital cellular pathways. In Fig. 1d, either CUL4A or DDB1 silencing increased the Sox2 protein level markedly, which further confirmed the role of CUL4A E3 ligase complex on ubiquitylating Sox2 for degradation.

2. Most of WB signals show clear difference between control and manipulation, but some of them are not clear enough if the results support the authors conclusion. It would be a good idea to quantify the signal intensities of WB in Fig 1h, Fig 2g, Fig 3d, Fig 4b.

Response: We thank the reviewer for the great suggestion. In the revised manuscript, we quantified the signal intensities of WB in Fig 1h (revised Fig. 1h), Fig 2g (revised Fig. 3c), Fig 3d (revised Fig. 4d) and Fig 4b (revised Fig. 6b) according to the comments.

3. It may be interesting to compare the temporal dynamics of Sox2 ubiquitination/degradation between NPCs and ESCs, to explain/discuss why those cells need to use the different pathway to control the levels of Sox2, how those dynamics could contribute the maintenance of ESCs and NPCs, respectively.

Response: We thank the reviewer for raising this interesting comment. To address this question, we firstly analyzed the half-life of Sox2 in ESCs, NPCs and neurons. The

result showed that the half-life of Sox2 protein in ESCs was longer than that in NPCs and neurons (revised Supplementary Fig. 8c). Then we also examined the protein levels of endogenous WWP2, Ube2Ss, Set7, COP1, DET1, CUL4A and, OTUD7B in ESCs, NPCs and neurons respectively. WWP2, Set7 and Ube2S had been reported to be the regulators of Sox2 proteostasis and ESCs maintenance. Our results showed that upon the differentiation from ESCs to NPCs, the level of WWP2 was decreased markedly, while the levels of Set7 and Ube2S were increased (revised Supplementary Fig. 1c). It is worth noting that WWP2 is hardly detectable in NPCs and Neurons. Consistently, WWP2 knockdown had minimal influence on half-life and ubiquitination of Sox2 in NPCs (revised Supplementary Fig. 4b and c), suggesting that WWP2 is not the major E3 for Sox2 in NPCs. Additionally, we found that during differentiation from ESCs to NPCs, there is a little change in the expression level of DET1, COP1 and CUL4A and they are at low level, while during differentiation from NPCs to neurons, their protein levels are increased gradually, suggesting that CUL4A^{DET1-COP1} E3 complex regulates Sox2 stability during the differentiation of NPC, but not that of ESC (revised Supplementary (Fig. 8d). Based on these observations, we speculate that different E3s control Sox2 stability/abundance at different differentiation stages.

Interestingly, we noted that OTUD7B was highly expressed in ESCs, moderately in NPCs, and lowly in neurons (Supplementary Fig. 8d), implying that OTUD7B might maintain Sox2 abundance in both ESCs and NPCs and it might be a pivotal deubiquitylase for Sox2. In this regard, the intact functions of OTUD7B in stem cells is worthy investigating in future studies.

4. Please quantify all blots and calculate differences using a relevant statistical test and report the p-values. This should be done for all comparisons not limited to the ones mentioned below, but with special attention to the ones below:

- **Fig. 1c, please quantify, calculate the correlation coefficient and p-value (corrected for multi-testing across all of the culins) for each comparison.**
- **Fig. 4a, it is difficult to assess the increase in Sox2, please quantify and add p-value for the test.**
- **Fig. 4c, it is difficult to assess the differences between the pattern with an sh for OTUD7A and OTUD7B, please quantify and report the p-value.**

Response: We thank the reviewer for the good suggestion. In the revised manuscript, we quantify all the blots and calculate the difference using a relevant statistical test as indicated in the Methods section and report the p-values.

For Fig. 1c, we show the correlation coefficient and p-value for Sox2 and all examined cullins as follow. And for Fig. 4a (revised Fig. 6a) and Fig. 4c (revised Fig. 6c), the correlation coefficient and p-value were shown in the revised figures.

Figure. The correlation coefficient and p-value for Sox2 and all examined Cullins in Fig. 1c of the manuscript.

5. In Fig 1d, knockdown of the adaptor DDB1 also increased Sox2 levels, please discuss the implications within the relevant results section.

Response: We thank the reviewer for this insightful suggestion. We rewrote the relevant results section and add the words as follow to discuss the implication of DDB1 regulating Sox2.

“CUL4 differs from other cullins in that it employs the WD40-like repeat-containing protein DDB1 as its adaptor to link CUL4 and DCAF. DDB1 had been documented to regulate some of vital cellular pathways as an integral component of the CUL4 CRL. To further confirm the effect of CUL4A E3 complex in NPCs, we examine Sox2 level in cells with knockdown of CUL1, CUL2, CUL3, CUL4A or DDB1. The results as shown in Fig. 1d showed that either CUL4A or DDB1 knockdown increased the Sox2 protein level markedly. These results underscored our speculation on CUL4A E3 ligase complex ubiquitylating Sox2 for degradation. Additionally, this finding suggested that DDB1 deficiency might inhibit NPC differentiation and lead to abnormal development of the nervous system. Consistently, Cang Y *et al* reported that null mutation of DDB1 caused early embryonic lethality, while conditional inactivation of DDB1 in brain and lens led to neuronal and lens degeneration, brain hemorrhages, and neonatal death [Cang Y *et al*, 2006]. So the regulation of DDB1 on Sox2 might reveal a novel mechanism underlying DDB1 knockout leading to neuronal degeneration.”

6. There are numerous grammatical errors throughout the manuscript.

Response: We thank the reviewer for pointing out the errors. We checked and rewrote the manuscript and corrected the grammatical errors.

References:

[1] Zielke N, Kim KJ, Tran V, Shibutani ST, Bravo MJ, Nagarajan S, van Straaten M, Woods B, von Dassow G, Rottig C, Lehner CF, Grewal SS, Duronio RJ, Edgar BA. Control of Drosophila endocycles by E2F and CRL4 (CDT2). *Nature*, 480:123-7 (2011).

[2] Cang Y, Zhang J, Nicholas SA, Bastien J, Li B, Zhou P, Goff SP. Deletion of DDB1 in mouse brain and lens leads to p53-dependent elimination of proliferating cells. *Cell*,127: 929-40 (2006).

Reviewer #2 (Remarks to the Author):

Cui et al performed a study on the regulation of the SOX2 protein during the differentiation of neural progenitor cells by ubiquitylation pathways leading to proteosomal degradation. SOX2 is a key factor maintaining a progenitor state in many cell lineages and elucidating the multi-layered mechanisms regulating its activity is of general interest. The authors identify the ubiquitin ligases and de-ubiquitination enzymes involved in this process. They further perform a variety of biochemical and cell biological assays to establish CUL4A, DET1/COP1 and OTUD7B as main components controlling Sox2 ubiquitination and stability. Overall the study contains a large amount of work (with a focus on co-immunoprecipitations to map interactions and Ub conjugations and western blots/immunocytochemistry to monitor protein levels), which is mostly well executed. However, the manuscript is difficult to read for the lack of subheadings and structure. It is certainly useful to pinpoint the exact E3 ligase (COP1) and DUB (OTUD7B) enzymes as well as the conjugation sites and interaction domains mediating the interactions with Sox2 but such insights are mainly of relevance to researchers with a focus on the regulation of protein degradation pathways. To a more general audience it comes as no surprise that the Sox2 turnover is post-translationally regulated but the relevance to stem cell biology and neurogenesis is not immediately clear. The authors close their manuscript by stating that their insights could have a ‘significant impact on improving the treatment of neural degenerative diseases.’ To justify this claim it could be shown that modifications of Sox2 levels by chemical perturbations of the UB/DUB system could expand the utility of NPCs or of derived differentiated cell types. In its present form the manuscript appears to be better suited for a more general journal.

Response: We thank the reviewer very much for the insightful concerns. We rewrote the manuscript according to the author instructions of *Nature Communications* (the original version is transferred from *Nature Cell Biology*), and added subheadings in the Results section to make it more logical and easy to understand.

In this study, we report the deubiquitylase OTUD7B and the Ring-finger E3 ubiquitin ligase complex CUL4A^{DET1-COP1} act as oppositional counterparts to control Sox2 ubiquitylation and protein stability at the post-translational level. We showed that Sox2 interacted with the WD40 repeats of COP1, and bound to the ZF (zinc finger) domain of OTUD7B. Moreover, in the revised manuscript we further mapped the ubiquitylation site of Sox2 using a series of KR mutants, and the results showed that Sox2 was ubiquitylated by COP1 on multiple lysines including K58, K65, K80, K87, K95, K103, K109, K122, K124 and K245 (revised Supplementary Fig. 4a). These findings shed more light on the biological mechanism underlying Sox2 degradation.

To explore the relevance of COP1 and OTUD7B to stem cell biology and neurogenesis, we knockdown COP1 or OTUD7B in NPCs and examined the expression of TUJ1, nestin and Sox2. COP1 silencing suppressed NPCs differentiation marked by decreased TUJ1 level (revised Fig. 5b and d), while OTUD7B knockdown led to increased NPCs differentiation (revised Fig. 7e-g). Overexpression experiments confirmed these conclusions (revised Fig. 5e and 7h). Further, we performed rescue assays to determine if the effect of COP1 or OTUD7B is dependent on its regulation on Sox2. Initially, we knockdown Sox2 in NPCs with COP1 silencing, and overexpressed Sox2-AA in NPCs with COP1 overexpression, then examined the level of Sox2 and TUJ1. Combined knockdown of Sox2 with COP1 significantly blunted the decrease of TUJ1 induced by COP1 knockdown (revised Fig. 5d and Supplementary Fig. 5b). While simultaneous expression of ectopic Sox2-AA and COP1 reversed the increase of TUJ1 level induced by COP1 overexpression alone (revised Fig. 5e and Supplementary Fig. 5c), indicating that COP1 induced NPCs differentiation in a Sox2-dependent manner, at least partially. Meanwhile, we expressed ectopic Sox2-AA in NPCs with OTUD7B knockdown, and silenced Sox2 in NPCs with OTUD7B overexpression. Exogenous Sox2-AA expression suppressed TUJ1 increase induced by OTUD7B silencing (revised Fig. 7g and Supplementary Fig. 7b), while Sox2 knockdown blocked the effect of exogenous OTUD7B on cell differentiation in NPCs (revised Fig. 7h and Supplementary Fig. 7c), suggesting that OTUD7B also plays a role in NPCs differentiation in a Sox2-dependent manner.

In the revised manuscript, we amended the last sentence in Discussion section to “Collectively, this study has uncovered a crucial post-translational control mechanism to regulate the key stem cell transcriptional factor Sox2 for cell fate determination of NPCs which will pave the way to improving the treatment of neural degenerative diseases.” In addition, we tested the effect of modifications of Sox2 level on NPCs differentiation by chemical perturbations of the UB/DUB system. Given that the specific inhibitor targeting CUL4A^{DET1-COP1} or OTUD7B is unavailable, we used unspecific DUB inhibitors PR-619 [Wang *et al* 2015] (Cat#S7130, Selleck), b-AP-15 [Crowder *et al* 2016] (Cat#S4920, Selleck) and the inhibitor of NAE (NEDD8-activating enzyme) Pevonedistat (MLN4924) (Cat#S7109, Selleck) which had been demonstrated to suppress neddylation of Cullins, and inactivate Cullin-RING E3 ligases [Nawrocki *et al* 2012]. The results as follow showed that compared with DMSO, the treatment of MLN4924 promoted NPC maintenance, while PR-619 and b-AP-15 enhanced cell differentiation. These data suggest that the strategy targeting UB/DUB system might have potential benefit in the treatment of neural degeneration diseases.

Figure. The effects of MLN4924, PR-619 and b-AP-15 treatment on NPC differentiation. NPCs were treated with indicated concentration of MLN4924, PR-619 and b-AP-15 for 24 h, and immunofluorescence analysis was conducted to examine cell differentiation. Green, Sox2; red, NeuN, a biomarker for neurons; blue, hoechst to stain nuclei. Scale bar, 50 µm.

Specific comments:

1) Acronyms are at times confusing and used interchangeably which can be confusing.

Response: We thank the reviewer to point out these errors. We carefully checked all the acronyms throughout the text, and corrected them in the revised manuscript. Regards to the requirements of Author Instruction, we use abbreviations throughout the text, but give the full wording once at the first place of mention in the main text.

2) Line 84: Description for “Supplementary Fig. 1b” does not match to the figure.

Response: We thank the reviewer very much for pointing out this mistake. In the revised manuscript, we reorganized the main figures and supplementary figures, and

made a double check throughout the main text and ensured the description matched to the figures.

3) Line 137: Marking mistake - “(Fig. 3d,e)” should be “(Fig. 2d,e)”.

Response: We apologize for the mistake. In the revised manuscript, we reorganized the main figures and supplementary figures, and made a double check throughout the main text and ensured the description matched to the figures.

4) Line 141: Marking mistake- “(Fig. 3e)” should be “(Fig. 2e)”.

Response: We have corrected it.

5) Number of biological replicates should be indicated and the statistical tests used should be mentioned.

Response: In the Methods section and Figure legends of the revised manuscript, we added the description on the number of biological replicates and the statistical tests.

6) Differences in band intensities are sometimes not obvious and should be supported by quantifications. For example, in Fig.3e, the difference in Sox2 levels between “siRNA-NT” and “siRNA-Set7” does not appear to be significant.

Response: Thank the reviewer for the good suggestion. We quantified the band intensities in Fig. 3e (revised Fig. 4e) and additionally, we quantified almost all blots and calculated differences in the revised Figures.

7) Line 214-216: “Then we detected the interaction between Sox2 and OTUD7B at endogenous (Fig. 4e) and exogenous (Supplementary Fig. 4f) levels and mapped the Sox2-binding region of OTUD7B to the ZF (zinc finger) domain (Fig. 4f).” The evidence for the identification of Sox2-binding region of OTUD7B is not convincing. Since the ZF is a DNA/RNA domain, interactions could simply be mediated by DNA/RNA scaffolds rather than being direct. Interactions with purified domains could further strengthen this point.

Response: We thank the reviewer for raising the insightful concerns. ZF (zinc finger) domain is a small protein structural motif that is characterized by the coordination of one or more zinc ions (Zn^{2+}) in order to stabilize the fold. The vast majority typically function as interaction modules that bind to DNA, RNA, proteins or other molecules. Here, we show that Sox2 binds to the ZF domain of OTUD7B. In order to confirm this finding, in the revised manuscript we generated the constructs of GST-OTUD7B-WT, GST-OTUD7B-N, GST-OTUD7B-C, GST-6P-OTUD7B- Δ ZF

and GST-6P-OTUD7B-ZF to express GST-tagged truncates. Using recombinant GST-OTUD7B-WT protein and these purified domains (GST-OTUD7B-N, GST-OTUD7B-C, GST-OTUD7B- Δ ZF and GST-OTUD7B ZF), we carried out GST pull-down assays in the cell free system. As shown in the revised Fig. 6g, Sox2 interacted with OTUD7B-WT, OTUD7B-C and OTUD7B-ZF which contain ZF domain, while neither OTUD7B-N nor OTUD7B- Δ ZF bound to Sox2, indicating that it is the ZF domain of OTUD7B that mediates direct binding to Sox2.

8) The methods lack detail. For example, the sequences of the FGF4 promoter used for the luciferase assay are unclear as is the amount of plasmids used in the assays.

Response: We thank the reviewer for pointing out these inadequacies. In the revised manuscript, we added the detailed description on FGF4 promoter sequence and the information of plasmids used in the assay. In addition, we improved the whole Methods section and provided more details.

9) Abbreviations need to be expanded at least once, suppliers of reagents and plasmid IDs should be given.

Response: We thank the reviewer to give the good suggestion. In the revised manuscript, the abbreviations are used throughout the text, but the full wordings are given once in the Abstract or once at the first place of mention in the main text. Additionally, suppliers of reagents and plasmid IDs are provided in the Methods section.

References:

- [1] Wang X, D'Arcy P, Caulfield TR, Paulus A, Chitta K, Mohanty C, Gullbo J, Chanan-Khan A, Linder S. Synthesis and evaluation of derivatives of the proteasome deubiquitinase inhibitor b-AP15. *Chem Biol Drug Des*, 86: 1036-48 (2015).
- [2] Crowder RN, Dicker DT, El-Deiry WS. The Deubiquitinase Inhibitor PR-619 Sensitizes Normal Human Fibroblasts to Tumor Necrosis Factor-related Apoptosis-inducing Ligand (TRAIL)-mediated Cell Death. *J Biol Chem*, 291: 5960-70 (2016).
- [3] Nawrocki ST, Griffin P, Kelly KR, Carew JS. MLN4924: a novel first-in-class inhibitor of NEDD8-activating enzyme for cancer therapy. *Expert Opin Investig Drugs*, 21: 1563-73 (2012).

Reviewer #3 (Remarks to the Author):

This is a nice manuscript describing the control of Sox2 protein stability during NPCs differentiation. Through high-quality biochemical experiments, the authors conclude that the CRL4-DET1-COP1 ligase complex is responsible for timely Sox2 protein degradation. Additionally, the authors identify OTUD7B as the de-ubiquitinase for Sox2. Interestingly, the CUL4 complex together with COP1 increases in expression during NPCs differentiation, while OTUD7B decreases, and these factors positively correlates with NPCs differentiation and negatively correlates with Sox2 protein level. Deregulation of Sox2 protein level by shRNA targeting COP1 or OTUD7B impairs NPCs differentiation. Therefore, the manuscript provides an important post-translational regulatory step for NPCs differentiation in a CRL4-DET1-COP1- and OTUD7B-dependent manner.

Response: We thank the reviewer for the kind comments on the thoroughness of our manuscript and for recognizing the novelty of this study.

Major comments:

- Page 6. Lines 125-127. The references reported by the authors (#24,#25) identify a canonical COP1's degron that is '[D/E]-X(2,3)-V-P-[D/E]' and not just 'V-P' as reported by the authors for Sox2. Hence, the degron that they identify is not a canonical degron and their statement is incorrect. As such, the mapping for the degron can be considered preliminary. I would suggest to remove these data unless supported by more extensive evidence (e.g., mapping the interaction of Sox2 N-term and C-term truncation mutant with COP1; analyze of the contribution for the binding of V283/P284 and V303/P304 independently, and provide additional protein stability analysis of the mutants eventually identified).

Response: We fully agree with the reviewer's advice and thank the reviewer for raising the insightful concern. According to the definition of COP1's degron, the 'V-P' motif identified in Sox2 is not a canonical COP1's degron, and our statement is incorrect. In the revised manuscript, we modified the description on VP motif to noncanonical degron.

We also attempted to confirm the ability of the noncanonical degron of Sox2 binding to COP1 by extensive evidence in the revised manuscript. Using GST pull-down assays, we examined the interaction of COP1 and Sox2 or Sox2 VP mutants. As shown in the revised Fig. 2e, the mutation of V283A/P284A (here named A1) retained the binding ability of Sox2 to COP1, while the V303A/P304A mutant (here named A2) and the V283A/P284A/V303A/P304A mutant (here named AA) lost the ability to interact with COP1 in the cell free system. Consistently, ectopic COP1 expression downregulated the levels of wide type Sox2 and the A1 mutant but not

those of Sox2 A2 and AA mutants (revised Fig. 3f). Therefore, we conclude that COP1 recognizes Sox2 mainly via the V303/P304 degron.

- **Page 8. Lines 163-164. Fig. 3d shows Sox2 protein levels, not its half-life. Therefore, it is incorrect to state that Sox2 K119R is subject to degradation by COP1/DET1. The same holds true for Fig. 3e. Moreover, the authors already shown that WWP2 was not expressed in their experimental condition. Nevertheless, it would be important to evaluate the contribution of the ligase WWP2 in Sox2 degradation and compare it to CRL4-DET1-COP1-mediated degradation. Given that the authors show RNA expression of WWP2 in supplementary Fig. 1a, I suggest to test the half-life of Sox2 comparing different conditions to evaluate the contribution of each ligase complex to the stability of Sox2 protein:**

1- after depletion of WWP2

2- after depletion of COP1 (and/or DET1)

3- after combined depletion of WWP2 and COP1 (or DET1)

Response: We thank the reviewer for pointing out the incorrect statements and bringing up the constructive suggestion. In the revised manuscript, we modified the incorrect statements mentioned by reviewer to “the mutation of K119R blunted Sox2 decrease induced by WWP2 overexpression. However, in COP1/DET1 or CUL4A overexpressed cells, this mutation had no influence on the Sox2 level.” and “siRNA targeting Set7 reversed the decrease of Sox2 level mediated by WWP2 overexpression. However, in COP1/DET1 or CUL4A overexpressed cells, Set7 knockdown had no such effect on Sox2 level”.

According to the comment, we generated NPCs with depletion of either WWP2, or COP1, or combined depletion of WWP2 and COP1, then the half-life and the ubiquitylation of Sox2 were determined to evaluate the contribution of WWP2 and COP1 to the stability of Sox2 protein. As shown in the revised Supplementary Fig. 4b and c, in NPCs with COP1 deletion or combined COP1 and WWP2 deletion, the half-life of Sox2 was prolonged and Sox2 ubiquitylation was decreased. In WWP2 deleted NPCs, the half-life and ubiquitylation of Sox2 were unchanged compared with the control cells. Thus, COP1-mediated ubiquitylation plays the major role in Sox2 degradation in NPCs.

- **Minor Comments:**

- **Page 4. 'Suppl. Fig. 1b', it seems that the authors are referring to 'Fig. 1c'.**

Response: We are so sorry for the mistake and thank the reviewer for pointing out that. In the revised manuscript, we reorganized the figures, and made a double check throughout the main text and ensured the description match to the figures.

- Fig. 2d. The order of the mutant and their labeling differ from the ordering and labeling in Fig. 2e. It would be better to change the order of the mutants in Fig. 2d (as the order of the same mutants in Fig. 2e) and to label Fig. 2e with the same labels used in Fig. 2d.

Response: We thank the reviewer to give the good advice for us. In the revised manuscript, we changed the labels of the mutants in the diagram and unified the labels of “WD40” and “R+CC” (Fig. 2d).

- Fig. 2h. The figure is mislabeled (the labeling is shifted down and covered by the first panel), hence, of difficult interpretation/evaluation.

Response: We are so sorry for the error and thank the reviewer for pointing out it. In the revised Fig. 3d, we relabeled the figure and make sure them correct.

- Page 7. Line 141. 'Fig. 3e', it seems that the authors are referring to 'Fig.2e'.

Response: We are so sorry for the error and thank the reviewer for pointing out it. In the revised manuscript, we rearranged the figures, and made a double check throughout the main text and ensured the description match to the figures.

- Page 8. Line 176. '...COP1 decreased gradually', it seems that the authors meant: '...COP1 increased gradually'.

Response: We thank the reviewer to correct this mistake in writing. In the revised manuscript, we amended the description as reviewer mentioned.

- Fig. 3f. The figure shows the juxtaposition of 3 channels. The readings of each separate channel should be reported at least in supplementary data.

Response: We thank the reviewer for raising the good suggestion. In the revised manuscript, we showed the images of each separate channel in main figures.

- Fig 4i. Why there are 4 lanes in Fig. 4i and only the last 2 lanes show a difference? Moreover, the levels of Sox2 are not shown.

Response: We thank the reviewer for pointing out the deficiency. We repeated this assay using MEFs from two mice with OTUD7B KO and two with OTUD7B WT, and showed better results in the revised Fig. 7c. Sox2 level was indicated in the revised figure. We also added the labels of mice number in revised Fig. 7a.

- Fig 4l. Labeling is not properly aligned.

Response: We thank the reviewer for pointing out the errors. In the revised manuscript, we relabeled the figure and ensured the labelings are properly aligned (revised Fig. 7f).

- Fig. 5b. The figure is shows the juxtaposition of 3 channels. The readings of each separate channel should be reported at least in supplementary data.

Response: We thank the reviewer for raising the good suggestion. In the revised manuscript, we showed the images of each separate channel in main figures.

Reviewers' Comments:

Reviewer #1:

Remarks to the Author:

I am satisfied with the responses to our concerns and recommend publication

Reviewer #2:

Remarks to the Author:

The authors have carefully revised the manuscript and added a substantial amount of additional data. Clarity of figures and readability of the results section has markedly improved. I am now supportive of publication in Nature Communications. I have a few remaining questions/comments.

- Figure 1a: error bars are not visible although legends states data are mean+/- s.d.
- Figure 1d/e/f: 'NC' refers to 'no CUL' or no construct control? Please explain in legend.
- Figure 2c: a His pulldown is shown in the Figure but not mentioned in the legend. Is there an extra panel which hasn't be described in the legend?
- I am somewhat surprised how 'clean' the co-IPs in this manuscript look when, for example, His-Sox1 and His-Sox2 are compared. These interactions must be really specific in order to have no background at all even for these highly homologous SoxB1 factors.
- As mouse as well as human cells were used it would better to state the species in all figure legends. I believe all experiments done with NPCs are for human cells. Is the experiment shown with the CUL4A/B knock-out MEFs the only one performed with mouse cells? In the legend of figure 5a the antibody is referred to anti-Sox2 which could imply a mouse protein (small cab).
- Page 13/14: Explain why the OTU family was selected out of the 98 DUBs present in the human genome. It's currently unclear why this family had been prioritised.
- Experiments in MEFs (i.e. Figures 1h/I and 7a-c): I am surprised as to the high expression of Sox2 in MEFs, As SOX2 is one of the four factors used to make iPSCs I would have expected SOX2 to be much lower in MEFs. Please comment on the protein expression levels of SOX2 in MEFs, NPCs and PSCs (pluripotent stem cells) and cite relevant literature that back up the observation as to the presence of SOX2 in MEFs. In our in-house RNAseq data for OG2-MEFs reads for endogenous SOX2 are barely above background levels
- The source (JAX ID of mice?) of the knock-out MEFs should be given (OTUD7B, CUL4A, CUL4B).

Reviewer #3:

Remarks to the Author:

The authors have thoroughly addressed my concerns

RESPONSE TO REVIEWERS' COMMENTS:

Reviewer #1 (Remarks to the Author):

I am satisfied with the responses to our concerns and recommend publication

Response: We sincerely thank the reviewer for the kind comments and the recommendation for publication.

Reviewer #2 (Remarks to the Author):

The authors have carefully revised the manuscript and added a substantial amount of additional data. Clarity of figures and readability of the results section has markedly improved. I am now supportive of publication in Nature Communications. I have a few remaining questions/comments.

Response: We thank very much the reviewer for the kind concerns and the recommendation for publication. According to the further comments, we have revised the manuscript and the responses are listed below point by point.

- **Figure 1a: error bars are not visible although legends states data are mean+/- s.d.**

Response: We thank reviewer for the comment. In the revised Figure 1a, we added the error bar which had been missed in previous version.

- **Figure 1d/e/f: 'NC' refers to 'no CUL' or no construct control? Please explain in legend.**

Response: We thank the reviewer for raising the kind concern. In these experiments, NC refers to empty vector control. We have added relevant description in the revised figure legends.

- **Figure 2c: a His pulldown is shown in the Figure but not mentioned in the legend. Is there an extra panel which hasn't be described in the legend?**

Response: We thank the reviewer for pointing out the missing. We have added the description on His pulldown in the legend in the revised manuscript. Additionally, we

rechecked all the figures and legends, and ensure that no extra panel which hasn't be described in the legend.

- **I am somewhat surprised how 'clean' the co-IPs in this manuscript look when, for example, His-Sox1 and His-Sox2 are compared. These interactions must be really specific in order to have no background at all even for these highly homologous SoxB1 factors.**

Response: We thank the reviewer for the concern. As the reviewer mentioned, almost no interaction between His-Sox1 and Flag-COP1 is observed in HEK293 cells. We repeated the Co-IP assay three times, and obtained the similar results. We further compared the amino acid sequences of Sox1 and Sox2, and we found that Sox1 lacks effective COP1-binding domain (canonical or noncanonical degrons) in spite of the high homology between Sox1 and Sox2, which might result in the marked difference of binding ability to COP1 between Sox1 and Sox2.

- **As mouse as well as human cells were used it would better to state the species in all figure legends. I believe all experiments done with NPCs are for human cells. Is the experiment shown with the CUL4A/B knock-out MEFs the only one performed with mouse cells? In the legend of figure 5a the antibody is referred to anti-Sox2 which could imply a mouse protein (small cab).**

Response: We thank the reviewer for the kind comments. In this manuscript, all experiments done with NPCs are for human cells and the experiments performed with MEFs from CUL4A/B or OTUD7B knock-out mice are for mouse cells. We have used 3 kinds of Sox2 antibody (#3579 (CST); sc-365823 and MAB2018) for immunoblot or immunofluorescence assay. Mouse Sox2 was immunoblotted with the antibody sc-365823 as described in the Methods section.

- **Page 13/14: Explain why the OTU family was selected out of the 98 DUBs present in the human genome. It's currently unclear why this family had been prioritised.**

Response: We thank the reviewer for raising the critical concerns. In our manuscript, we screened the specific deubiquitylas (DUB) for Sox2 from OTU family. OTU

DUBs regulate important cell signalling pathways. For example, OTUD5/DUBA regulates IRF3 signalling ^[1]; A20 regulates NF-κB signaling ^[2]; OTUB1 regulates the DNA damage response ^[3]; OTUD7B controls non-canonical NF-κB activation in immune regulation ^[4] and OTULIN regulates linear ubiquitylation and innate immune signalling as well as angio genesis ^[5-7]. Additionally, several OTU DUBs were reported to be ubiquitin (Ub) chain linkage specific ^[8]. Regards to importance of OTUs in cell-signaling cascades and the specificity for regulating Ub chains linkage, we prioritized the OTUs family as the candidates for screening DUB for Sox2.

Reference:

1. Kayagaki, N. et al. DUBA: a deubiquitinase that regulates type I interferon production. *Science* 318, 1628–1632 (2007).
2. Hymowitz, S. G. & Wertz, I. E. A20: from ubiquitin editing to tumour suppression. *Nat. Rev. Cancer* 10, 332–341 (2010).
3. Nakada, S. et al. Non-canonical inhibition of DNA damage-dependent ubiquitination OTUB1. *Nature* 466, 941–946 (2010).
4. Hu, H. et al. OTUD7B controls non-canonical NF-κB activation through deubiquitination of TRAF3. *Nature* 494, 371–374 (2013).
5. Keusekotten, K. et al. OTULIN antagonizes LUBAC signaling by specifically hydrolyzing Met1-linked polyubiquitin. *Cell* 153, 1312–1326 (2013).
6. Fiil, B. K. et al. OTULIN restricts Met1-linked ubiquitination to control innate immune signaling. *Mol. Cell* 50, 818–830 (2013).
7. Rivkin, E. et al. The linear ubiquitin-specific deubiquitinase gumby regulates angiogenesis. *Nature* 498, 318–324 (2013).
8. Tycho E.T. Mevissen, Manuela K. Hospenthal, Paul P. Geurink, et al. OTU Deubiquitinases Reveal Mechanisms of Linkage Specificity and Enable Ubiquitin Chain Restriction Analysis. *Cell* 154, 169–184 (2013).

- **Experiments in MEFs (i.e. Figures 1h/I and 7a-c): I am surprised as to the high expression of Sox2 in MEFs, As SOX2 is one of the four factors used to make iPSCs I would have expected SOX2 to be much lower in MEFs. Please comment on the protein expression levels of SOX2 in MEFs, NPCs and PSCs**

(pluripotent stem cells) and cite relevant literature that back up the observation as to the presence of SOX2 in MEFs. In our in-house RNAseq data for OG2-MEFs reads for endogenous SOX2 are barely above background levels.

Response: We thank the reviewer for bringing up the important concern and highly appreciated the reviewer's stringent analysis for this point. As the reviewer expected, Sox2 protein level in MEF is indeed much lower than ESC or NPCs. In order to present the Sox2 expression clearly, we used the SuperLumina ECL Substrate Kit (ThermoFisher) to develop the immunoblotted membrane in our experiment, which increased the intensity of Sox2 immunoblot.

- **The source (JAX ID of mice?) of the knock-out MEFs should be given (OTUD7B, CUL4A, CUL4B).**

Response: We thank the reviewer for giving the good suggestion. We obtained the CUL4A/B knock-out MEFs from Dr. Pengbo Zhou (Weill Cornell Medical College). OTUD7B knock-out MEFs were purchased from Model Animal Research Center of Nanjing University, China. We added the relevant description in the Methods section.

Reviewer #3 (Remarks to the Author):

The authors have thoroughly addressed my concerns

Response: We sincerely thank the reviewer for the kind comments and the recommendation for publication.